# AN EFFICIENT, PROVABLY OPTIMAL ALGORITHM FOR THE 0-1 LOSS LINEAR CLASSIFICATION PROBLEM

**Xi He**
School of Computer Science
Peking University
Beijing, China
xihe@pku.edu.cn

**Max A. Little**
School of Computational Neuroscience
University of Birmingham
Birmingham, UK
maxl@mit.edu

## ABSTRACT

Algorithms for solving the linear classification problem have a long history, dating back at least to 1936 with linear discriminant analysis. For linearly separable data, many algorithms can obtain the exact solution to the corresponding 0-1 loss classification problem efficiently, but for data which is not linearly separable, it has been shown that this problem, in full generality, is NP-hard. Alternative approaches all involve approximations of some kind, such as the use of surrogates for the 0-1 loss (for example, the hinge or logistic loss), none of which can be guaranteed to solve the problem exactly. Finding an efficient, rigorously proven algorithm for obtaining an exact (i.e., globally optimal) solution to the 0-1 loss linear classification problem remains an open problem.

By analyzing the combinatorial and incidence relations between hyperplanes and data points, we derive a rigorous construction algorithm, incremental cell enumeration (ICE), that can solve the 0-1 loss classification problem exactly in $O\left(N^{D+1}\right)$—exponential in the data dimension $D$. To the best of our knowledge, this is the first standalone algorithm—one that does not rely on general-purpose solvers—with rigorously proven guarantees for this problem. Moreover, we further generalize ICE to address the polynomial hypersurface classification problem in $O\left(N^{G+1}\right)$ time, where $G$ is determined by both the data dimension $D$ and the polynomial degree $K$ defining the hypersurface. The correctness of our algorithm is proved by the use of tools from the theory of hyperplane arrangements and oriented matroids.

We demonstrate the effectiveness of our algorithm on real-world datasets, achieving optimal training accuracy for small-scale datasets and higher test accuracy on most datasets. Furthermore, our complexity analysis shows that the ICE algorithm offers superior computational efficiency compared with state-of-the-art branch-and-bound algorithm.

## 1 INTRODUCTION

Increasingly, machine learning (ML) is being used for high-stakes prediction applications that deeply impact human lives. Many of these ML models are "black boxes" with highly complex, inscrutable functional forms. In high-stakes applications such as healthcare and criminal justice, black box ML predictions have incorrectly denied parole (Wexler, 2017), misclassified highly polluted air as safe to breathe (McGough, 2018), and suggested poor allocation of valuable, limited resources in medicine and energy reliability (Varshney & Alemzadeh, 2017). In such high-stakes applications of ML, we always want the best possible prediction, and we want to know how the model makes these predictions so that we can be confident the predictions are meaningful (Rudin, 2022). In short, the ideal model is simple enough to be easily understood (*interpretable*), and optimally accurate (*exact*).

Another compelling reason why simple models are preferable is because such *low complexity* models usually provide better *statistical generality*, in the sense that a classifier fit to some training dataset, will work well on another dataset drawn from the same distribution to which we do not have access (works well *out-of-sample*). The *VC dimension* is a key measure of the complexity of a classification

model. The simple $D$-dimensional *linear hyperplane* classification model, which we discuss in detail below, has VC dimension $D + 1$ which is the lowest of other widely used models such as the decision tree model (axis-parallel hyper-rectangles, VC dimension $2D$), the $K$-degree polynomial (VC dimension $O\left(D^K\right)$) and the $L$-layer, $W$-weight piecewise linear deep neural networks (VC dimension $O\left(WL\log\left(W\right)\right)$), for instance (Vapnik, 1999; Blumer et al., 1989; Bartlett et al., 2019).

Assume a dataset of size $N$ is drawn i.i.d (independent and identically distributed) from the same distribution as the training dataset, according to Vapnik (1999)'s *generalization bound theorem*, for the hyperplane classifier we have, with high probability,

$$E_{\text{test}} \leq E_{\text{emp}} + O\left(\sqrt{\frac{\log\left(N/\left(D+1\right)\right)}{N/\left(D+1\right)}}\right), \tag{1}$$

where $E_{\text{test}}$, $E_{\text{emp}}$ are the *test 0-1 loss* and *empirical 0-1 loss* of on training dataset, respectively (Mohri et al., 2018). Equation (1) motivates finding the exact (gloablly optimal) 0-1 loss on the training data and simplest model, as the lower the training accuracy and the model complexity (defined by VC-dimension) the more likely the model will obtain a better result on testing dataset. If a data set is simple enough, a linear classifier can deliver an accurate enough solution. In which case, no other model can outperform the exact linear classifier.

Training a model to global optimality on a training dataset is known as the empirical risk minimization problem. However, even for perhaps the simplest case—the linear model—training a model to global optimality is intractable. It has long been proven that empirical risk minimization for 0-1 loss (i.e., minimizing the number of misclassifications) in linear classification is NP-hard (Ben-David et al., 2003) as a function of the data dimension (Mohri et al., 2018).

Consequently, most algorithms proposed for this problem focus on optimizing approximate variants of the 0-1 loss, such as the logistic loss (Cox, 1958; 1966), and hinge loss (Cortes & Vapnik, 1995). By contrast, relatively little attention has been given to exact algorithms for the 0-1 loss classification problem (0-1 LCP). One approach is to formulate the problem as a mixed-integer program (MIP) and solve it using general-purpose solvers, such as Gurobi (Gurobi Optimization, LLC, 2024). For instance, Tang et al. (2014) employed a MIP formulation to obtain the maximum-margin boundary under 0-1 loss, while Brooks (2011) optimized the "ramp loss" and the hard-margin loss—both closely related to 0-1 loss—using a quadratic mixed-integer program (QMIP).

Alternatively, combinatorial methods such as the branch-and-bound (BnB) approach have also been applied. Nguyen & Sanner (2013) for example, proposed several BnB-based algorithms for solving the 0-1 LCP. However, a common problem in BnB research is the lack of formal proofs of exhaustiveness, making the correctness of such algorithms uncertain. Although Nguyen & Sanner (2013) present several interesting methods, none of them are accompanied with a formal correctness proof.

Nevertheless, the well-known Cover's functional counting theorem (Cover, 1965) rigorously established that there are

$$\text{Cover}\left(N, D+1\right) = 2\sum_{d=0}^{D}\left(\begin{array}{c} N-1 \\ d \end{array}\right) = O\left(N^D\right) \tag{2}$$

possible *linear dichotomies* of $N$ points in $\mathbb{R}^D$. This result suggests that, in principle, one could solve the 0-1 LCP exactly by exhaustively enumerating these partitions. However, Cover's result is purely combinatorial and does not provide any method for performing this enumeration.

Interestingly, Nguyen & Sanner (2013) observed that selecting hyperplanes formed by choosing $D$ out of $N$ data samples suffices to solve the 0-1 loss LCP exactly. This procedure has a combinatorial complexity of $\left(\begin{array}{c} N \\ D \end{array}\right)$, which appears to be smaller than the complexity derived from Cover's analysis. At the same time, in the context of the hyperplane decision tree problem, Murthy et al. (1994); Dunn (2018) observed that all possible linear partitions can be enumerated in $2^D\left(\begin{array}{c} N \\ D \end{array}\right)$, which is larger than the bound implied by Cover's result. These three distinct combinatorial analyses yield seemingly inconsistent complexity estimates. This naturally raises the question:

*Which of these analyses is correct for solving the 0-1 loss linear classification problem? If all are valid, how are they connected?*

This paper is dedicated to addressing these questions formally. Our key contributions are as follows:

- **Combinatorial foundations for classification in Euclidean space**: We establish the combinatorial and incidence relationships between hyperplane arrangements and point configurations in the ordinary vector space $\mathbb{R}^D$. Unlike the classical treatment in combinatorial geometry and oriented matroid theory—which is based on homogeneous coordinates—we work directly in inhomogeneous (Euclidean) coordinates[1] (Edelsbrunner, 1987; Fukuda, 2016).

- **A novel 0-1 loss linear classification theorem**: We present a new Theorem 3 for solving the 0-1 LCP, which rigorously proves why Nguyen & Sanner (2013)'s prioritized combinatorial search (PCS) algorithm can exactly solve the 0-1 LCP. The supporting lemmas of Theorem 3 reveal deep connections between the three distinct combinatorial analyses Cover (1965), Murthy et al. (1994); Dunn (2018), and Nguyen & Sanner (2013).

- **The first rigorously proven standalone algorithm for 0-1 linear classification problem**: By combining Theorem 3 with the efficient combination generator introduced by He & Little (2025), we construct the first rigorously proven, standalone algorithm—one that does not rely on general-purpose solvers—for solving the 0-1 LCP. Empirical results (see Figure 5) show that, for example, when $N = 150$ data size with $D = 3$, ICE would take **1.2 seconds** worst-case whereas Nguyen and Sanner (2013)'s BnB would take approximately $10^{10}$ seconds (nearly **317 years**), clearly demonstrating the superiority of our approach.

- **Extension to polynomial hypersurface classification**: We extend our theoretical framework to polynomial hypersurfaces, resulting in an optimal algorithm for solving the 0-1 loss hypersurface classification problem.

- **Empirical insights on generalization**: Our experiments show that solutions with lower training accuracy often generalize better to unseen test data. This observation refutes the conventional belief that exact algorithms overfit and is consistent with Vapnik (1999)'s generalization bound theorem.

The paper is organized as follows. In Section 2, we provide a detailed geometric analysis of the linear classification problem and develop novel theorems for solving the 0-1 loss linear and hypersurface classification problems. This result leads to a new class of algorithms capable of solving these problems exactly. Section 3 presents empirical results comparing the ICE algorithm with standard approximate methods on real-world datasets from the UCI Machine Learning Repository, evaluating both training accuracy and out-of-sample generalization performance. Finally, Section 4, discusses our contributions and the limitations of the proposed algorithm, and outlines potential directions for future research.

## 2 THEORY

### 2.1 PROBLEM DEFINITION

Assume a dataset consists of $N$ *data points* (or data items) $\boldsymbol{x}_n, \forall n \in \{1, \ldots, N\} = \mathcal{N}$, where the data points $\boldsymbol{x}_n \in \mathbb{R}^D$ and $D$ is the dimension of the *feature space*. Each data point has a unique true *label* $l_n \in \{-1, 1\}, \forall n \in \mathcal{N}$. All true labels in this dataset are stored in set $\boldsymbol{l} = \{l_1, l_2, ..., l_N\}$. The data points and their labels are packaged together into the dataset $\mathcal{D}$, denoted as $\mathcal{D}_{\boldsymbol{l}}$. The 0-1 LCP can be defined as

$$\hat{\boldsymbol{w}} = \operatorname*{argmin}_{\boldsymbol{w} \in \mathbb{R}^{D+1}} E_{\text{0-1}}\left(\boldsymbol{w}, \mathcal{D}_{\boldsymbol{l}}\right) = \sum_{n \in \mathcal{N}} \mathbf{1}\left[\operatorname{sign}\left(\boldsymbol{w}^T \bar{\boldsymbol{x}}_n\right) \neq l_n\right]. \tag{3}$$

where $E_{\text{0-1}}\left(\boldsymbol{w}, \mathcal{D}_{\boldsymbol{l}}\right) = \sum_{n \in \mathcal{N}} \mathbf{1}\left[\operatorname{sign}\left(\boldsymbol{w}^T \bar{\boldsymbol{x}}_n\right) \neq l_n\right]$ is the *0-1 loss objective function* which counts the number of misclassified data points given the parameter $\boldsymbol{w}$, we denote $E_{\text{0-1}}\left(\boldsymbol{w}, \mathcal{D}_{\boldsymbol{l}}\right)$ as $E_{\text{0-1}}\left(\boldsymbol{w}\right)$

---

[1]Informally a dataset in $\mathbb{R}^D$ is in inhomogeneous coordinates, whereas $\bar{\boldsymbol{x}} = (\boldsymbol{x}, 1)$ represents the same point in homogeneous coordinates (projective space)

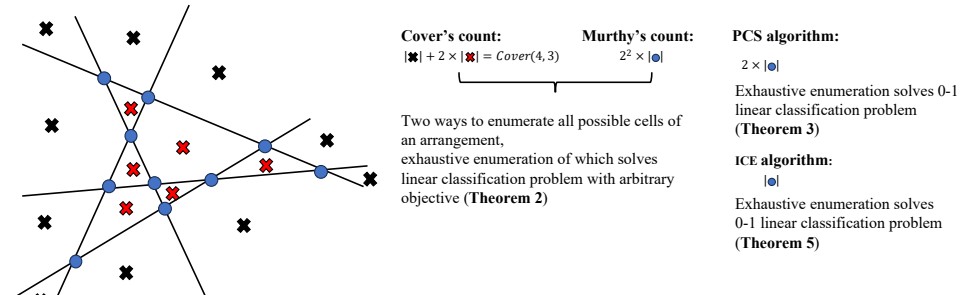

Figure 1: Novel theoretical contributions enabling the ICE algorithm: identifying the necessary and sufficient dual-arrangement faces that must be enumerated to solve the 0-1 LCP. The black $\times$ marks (unbounded cells) and red $\times$ marks (bounded cells) represent all the cells of a dual arrangement, with $|\cdot|$ denoting their size. In Theorem 2, we show that exhaustively enumerating all cells and the reversals of unbounded cells (with total size $|\times| + 2\,|\times|$) yields a number exactly matching Cover's counting function $Cover$ for possible linear dichotomies (as proved in Lemma 1). This procedure solves the linear classification problem for any objective function, filling the gap in Cover's theorem, which provides only a counting formula without specifying how to enumerate the dichotomies. Theorem 3 demonstrates that the 0–1 LCP can be solved exactly by exhaustively enumerating all *blue circles* in the figure and their corresponding reversed sign vectors, formally proving the correctness of Nguyen & Sanner (2013)'s PCS algorithm, which had only been **empirically observed** to be optimal. Finally, Theorem 5 shows that it suffices to enumerate only the blue circles, without their reversed signs, reducing the number of configurations and enabling the construction of our *incremental cell enumeration* (ICE) algorithm.

when $\mathcal{D}_\mathbf{l}$ is clear from the context. The supervised classification problem is solved by computing (3) which is a sum of 0-1 loss functions $\mathbf{1}[\ ]$, each taking the value 1 if the Boolean argument is true, and 0 if false. The function sign returns $+1$ is the argument is positive, and $-1$ if negative (and zero otherwise). The linear decision function $\boldsymbol{w}^T\bar{\boldsymbol{x}}$ with parameters $\boldsymbol{w} \in \mathbb{R}^{D+1}$ and $\bar{\boldsymbol{x}} = (\boldsymbol{x}, 1)$ is a data point in *homogeneous coordinates*. Although apparently simple, this is a surprisingly challenging optimization problem. Considered as a continuous optimization problem, the standard ML optimization technique, gradient descent, is not applicable (since the gradients of $E_{0\text{-}1}$ with respect to $w$ are zero everywhere they exist), and the problem is non-convex so there are a potentially very large number of local minima in which gradient descent can become trapped. Nevertheless, the finiteness of the dataset implies that only a finite number of partitions are possible. In particular, we are concerned with those partitions that can be induced by hyperplanes—i.e., linear dichotomies. The next subsection explains how to identify these linear dichotomies using a geometric dual transformation, which can then be applied to solve (3).

A diagrammatic summary of the key geometric results is presented in Figure 1.

## 2.2 POINT CONFIGURATIONS AND HYPERPLANE ARRANGEMENTS

A *point configuration* is synonymous with a dataset and is denoted by $\mathcal{P} = \{\boldsymbol{p}_n \in \mathbb{R}^D : n \in \mathcal{N}\}$. A finite *hyperplane arrangement* is a finite set of hyperplanes $\mathcal{H} = \{h_1, ..., h_k\}$, where each hyperplane is defined as $h_n = \{\boldsymbol{x} \in \mathbb{R}^D : \boldsymbol{w}^T\boldsymbol{x} = c\}$ for some constant $c \in \mathbb{R}$. A point configuration or hyperplane arrangement in *general position* is called *simple* if no $k$ of them lie in a $(k-2)$-dimensional affine subspace of $\mathbb{R}^D$ and the intersection of any $k$ hyperplanes is contained in a $(D-k)$-dimensional *flat*, for $1 \le k \le D$. For example, if $D = 2$ then a set of lines is in general position if no two are parallel and no three meet at a point.

**Definition 1.** *Faces of a hyperplane arrangement*. Let $\mathcal{F}_\mathcal{H}$ be the set of all sign vectors $\text{sign}_\mathcal{H}(\boldsymbol{x})$ in $\mathbb{R}^D$ for arrangement $\mathcal{H}$, which is defined as

$$\mathcal{F}_\mathcal{H} = \left\{\text{sign}_\mathcal{H}(\boldsymbol{x}) : \boldsymbol{x} \in \mathbb{R}^D\right\}, \tag{4}$$

A *face $f$* (connected component) of an arrangement $f \subseteq \mathbb{R}^D$ is a maximal subset of $\mathbb{R}^D$, such that all $\boldsymbol{x} \in f$ have the same sign vector $\text{sign}_{\mathcal{H}}(\boldsymbol{x}) \in \mathcal{F}_{\mathcal{H}}$. Given a sign vector $\text{sign}_{\mathcal{H}}(\boldsymbol{x}) = (\delta_1(\boldsymbol{x}), \delta_2(\boldsymbol{x}), \ldots, \delta_I(\boldsymbol{x}))$, the connected region of $f$ can be defined as $f = \bigcap_{i \in \mathcal{I}} h_i^{\delta_i(f)}$ . In fact, $f$ defines an *equivalence class* in $\mathbb{R}^D$. Since any point $\boldsymbol{x} \in f$ has the same sign vector, then $\text{sign}_{\mathcal{H}}(f)$ is *the* sign vector for any point in $f$. A face is said to be *$k$-dimensional* if it is contained in a *$k$-flat* for $-1 \leq k \leq D + 1$. Some special faces are given specific names *vertices* ($k = 0$), *edges* ($k = 1$), and *cells* ($k = D$). A $k$-face $g$ and a $(k-1)$-face $f$ are said to be *incident* if $f$ is contained in the boundary of face $g$, for $1 \leq k \leq D$. In that case, face $g$ is called a *superface* of $f$, and $f$ is called a *subface* of $g$. The cells in an arrangement can be further split into two classes, the *bounded cells* and *unbounded cells*. Informally, a cell is bounded if it is a closed region surrounded by hyperplanes (the boundaries are not contained in cells), and unbounded otherwise.

Superficially, a hyperplane arrangement might seem to contain more information or structure than a set of data points (a point configuration). However, a valuable approach to studying geometric objects involving points and hyperplanes is to explore the transformations between these two objects. By studying the *dual transformation* between point configurations and hyperplane arrangements, it will later be seen that the superficial impression of the structural information contained in hyperplane arrangement and point configuration is incorrect. Both hyperplane arrangements and point configurations possess equally rich combinatorial structure.

In the next section, we examine the geometric relationships among points, hyperplanes, and dichotomies, with a focus on their combinatorial and incidence relations, leading to a new perspective on the linear classification problem. This enables the development of an efficient and general algorithm capable of solving linear classification problems. **Detailed proofs of all theorems and lemmas in next section are provided in the Appendix** A.

### 2.3 LINEAR CLASSIFICATION AND POINT-HYPERPLANE DUALITY

The geometric dual transformation $\phi : \mathbb{R}^D \rightarrow \mathbb{R}^D$ maps a point $\boldsymbol{p}$ to a non-vertical affine hyperplane $\phi(\boldsymbol{p})$, defined by the equation

$$p_1 x_1 + p_2 x_2 + \ldots + p_{D-1} x_{D-1} - x_D = p_D, \tag{5}$$

and conversely, the function $\phi^{-1}$ transforms a (non-vertical) hyperplane $h$ defined by polynomial $w_1 x_1 + w_2 x_2 + \ldots + w_{D-1} x_{D-1} - x_D = w_D$ to a point $\phi^{-1}(h) = (w_1, w_2, \ldots, w_D)^T$. The terms *primal space*, and *dual space* refer to the spaces before and after transformation by $\phi$ and $\phi^{-1}$. The dual transformation is naturally extended to a set of points $\phi(\mathcal{P})$ and a set of hyperplanes $\phi(\mathcal{H})$ by applying it to all points and hyperplanes in the set. We have the following important theorem which is the foundation for analysising the incidence and combinatorial relations between data points and linear dichotomies.

**Theorem 1.** *Incidence relations of the dual transformation.* Let $\boldsymbol{p}$ be a point and a non-vertical affine hyperplane $h = \{\boldsymbol{x} : \boldsymbol{w}^T \boldsymbol{x} = 0\}$ in $\mathbb{R}^D$. Under the dual transformation $\phi$, $\boldsymbol{p}$ and $H$ satisfy the following properties:

1. *Incidence preservation*: Point $\boldsymbol{p}$ belongs to hyperplane $h$ if and only if point $\phi^{-1}(h)$ belongs to hyperplane $\phi(\boldsymbol{p}) = p$,

2. *Order preservation*: Point $\boldsymbol{p}$ lies above (below) hyperplane $h$ if and only if point $\phi^{-1}(h)$ lies above (below) hyperplane $\phi(\boldsymbol{p})$.

That the dual transformation preserves the incidence relations above can be proved by examining the relationship between the dual transformation $\phi$ and the *unit paraboloid* (Edelsbrunner, 1987). The incidence preservation property described above implies a duality between the definitions of general position for point configurations and hyperplane arrangements. For instance, when $D = 2$, three points lying in the same 1-flat $l$ (a line) correspond to three lines in the dual space intersecting at the same point $\phi(l)$, these three lines are mutually parallel if the line $l$ is vertical.

It can be difficult to visualize how Cover's dichotomies form equivalence classes for decision hyperplanes, but the same decision hyperplanes in the dual space $\phi(\mathcal{P})$, $\forall \boldsymbol{p} \in \mathbb{R}^D$ partition the space into different cells, where each cell corresponds to an equivalence class of dichotomies (Fig. 2). Moreover, this explains why the prediction by Murthy et al. (1994); Dunn (2018), of $2^D \begin{pmatrix} N \\ D \end{pmatrix}$ possible

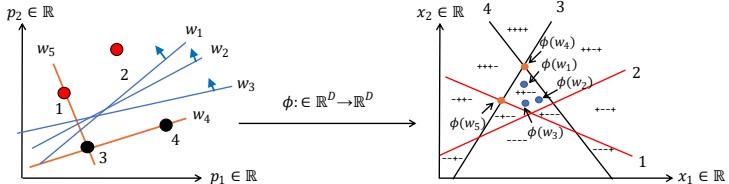

Figure 2: A point configuration $\mathcal{D}$ (left-panel) and its dual arrangement $\mathcal{H}_{\mathcal{D}}$ (right-panel). The yellow hyperplanes $w_4$, $w_5$ with two points lying on them in $\mathbb{R}^D$ correspond to the yellow points in the dual space, which are the intersection of corresponding dual hyperplanes $\phi(w_4)$, $\phi(w_5)$. For (blue) hyperplanes $w_1$, $w_2$, $w_3$ with the same prediction labels $(+, +, -, -)$, their corresponding dual points $\phi(w_1)$, $\phi(w_2)$, $\phi(w_2)$ lie in the same cell of dual arrangement $\phi(\mathcal{D})$.

linear classifications is correct: the most straightforward way to enumerate all cells in a hyperplane arrangement is to first enumerate the $\begin{pmatrix} N \\ D \end{pmatrix}$ vertices—each determined by a unique combination of $D$ hyperplanes in general position—and then consider the $2^D$ adjacent cells associated with each vertex. This enumeration relies on the general position assumption, which guarantees that every $D$-combination of hyperplanes defines a distinct vertex with exactly $2^D$ neighboring regions.

Importantly, we present the following lemma, which explains the combinatorial relationship between linear dichotomies and the cells of the dual arrangement. This lemma is the basis for an alternative approach to proving Cover's counting theorem.

**Lemma 1.** For a set points $\mathcal{D} = \left\{ \mathbf{x}_n \in \mathbb{R}^D : n \in \mathcal{N} \right\}$ in general position, the total number of linear dichotomies in Cover's function counting theorem, is the same as the number of cells of the dual arrangement $\mathcal{H}_{\mathcal{D}}$, plus the number of bounded cells of $\mathcal{H}_{\mathcal{D}}$. In other words, denote the number of dichotomies for $N$ data items in $\mathbb{R}^D$ as $\text{Cover}(N, D+1)$ $(D+1$ denote the dimension of data in homogeneous coordinates) and the number of cells and bounded cells of an hyperplane arrangement in $\mathbb{R}^D$ as $B_D(\mathcal{H}_{\mathcal{D}})$ and $C_D(\mathcal{H}_{\mathcal{D}})$. Then

$$\text{Cover}(N, D+1) = B_D(\mathcal{H}_{\mathcal{D}}) + C_D(\mathcal{H}_{\mathcal{D}}) \tag{6}$$

Another, previously reported, geometric analysis on the combinatorial relations between the hyperplane arrangement and the point configuration is based on *homogeneous coordinates*, where all cells of the dual arrangement are unbounded (Edelsbrunner, 1987; Fukuda, 2016).

The equivalence between the number of dichotomies and the sum of the number of bounded cells and the number of cells may initially seem unclear. The intuition lies in the fact that not every dichotomy in the primal space corresponds to a cell in the dual space. Specifically, decision boundaries associated with unbounded cells correspond to two dichotomies, whereas those associated with bounded cells correspond to only one. This relationship is clarified by the following lemma.

**Lemma 2.** For a dataset $\mathcal{D}$ in general position, each of Cover's dichotomies corresponds to a cell in the dual space, and dichotomies corresponding to bounded cells have no *complement cell* (cells with reverse sign vector). Dichotomies corresponding to the unbounded cells in the dual arrangements $\phi(\mathcal{D})$ have a complement cell.

Since each of Cover's dichotomies corresponds to a cell in the dual space, and dichotomies corresponding to bounded cells have no complement cell (cells with reverse sign vector), lemma 2 demonstrates that all possible *Cover's dichotomies* of a given dataset $\mathcal{D}$ can be obtained by enumerating the cells of an arrangement and the complemented cells of the bounded cells. The enumeration of the complements of the bounded cells requires an additional process, as the bounded cells within the arrangement do not have complementary cells. This result leads directly to the following theorem.

**Theorem 2.** *Linear classification theorem.* Let $\mathcal{D}$ be a data set in general position in $\mathbb{R}^D$. If an $O\left(N^{D+1}\right)$-time cell enumeration algorithm exists, then exact solutions for the linear classification problem with an arbitrary objective function can be obtained in at most $O\left(t_{\text{eval}} \times N^{D+1}\right)$ time by exhaustively enumerating the cells of the dual arrangement $\mathcal{H}_{\mathcal{D}}$, where $t_{\text{eval}}$ represents the time required to evaluate the classification objective.

Theorem 2 gives us a method for solving the linear classification problem over *arbitrary* objective function. However, as we interested in only the LCP with 0-1 loss objective (3), the properties below helps us to solve the LCP over 0-1 loss more efficiently. The next lemma explains not only that Cover's dichotomies have corresponding dual cells for the dual hyperplane arrangement, but also that hyperplanes containing $0 \leq k \leq D$ data points have corresponding dual faces.

**Lemma 3.** For a dataset $\mathcal{D}$ in general position, a hyperplane with $k$ data items lying on it, $0 \leq k \leq D$ correspond to a $(D - k)$-face in the dual arrangement $\mathcal{H}_{\mathcal{D}}$. Hyperplanes with $D$ points lying on it, correspond to vertices in the dual arrangement.

**Definition 2.** Given a hyperplane arrangement $\mathcal{H} = \{h_n : n \in \mathcal{N}\}$. The separation set $sep\,(f, g)$ for two faces $f$, $g$ is defined by

$$\text{sep}\,(f, g) = \{n \in \mathcal{N} : \delta_n\,(f) = -\delta_n\,(g) \neq 0\}, \tag{7}$$

using which, we say that the two faces $f$, $g$ are *conformal* if $\text{sep}\,(f, g) = \emptyset$.

That two faces that are conformal is essentially the same thing as saying that two faces have consistent classification assignments.

**Lemma 4.** Given a hyperplane arrangement $\mathcal{H} = \{h_n : n \in \mathcal{N}\}$, two faces $f$, $g$ are conformal if and only if $f$ and $g$ are subfaces of a common face or one face is a subface of the other.

A similar result is described in *oriented matroid* theory (Björner, 1999). The following lemma will be instrumental in the analysis, presented later, of the linear classification problem with the *0-1 loss* objective. It suggests that the optimal cell, with respect to 0-1 loss, is conformal to the optimal vertex.

**Lemma 5.** Given a hyperplane arrangement $\mathcal{H} = \{h_n : n \in \mathcal{N}\}$, for an arbitrary maximal face (cell) $f$, the sign vector of $f$ is $\text{sign}_{\mathcal{H}}\,(f)$. For an arbitrary $(D - d)$-dimension face $g$, $0 < d \leq D$, the number of different signs of $\text{sign}_{\mathcal{H}}\,(g)$ with respect to $\text{sign}_{\mathcal{H}}\,(f)$ is larger than or equal to $d$, where equality holds only when $g$ is conformal to $f$ ($g$ is a subface of $f$).

Now we have all receipts to prove the final result, for the linear classification problem over 0-1 loss, we can solve it by exhuastively searching all $D$-combinations of data points. The following theorem formally proves Nguyen & Sanner (2013)'s observation.

**Theorem 3.** *0-1 loss linear classification theorem.* Consider a dataset $\mathcal{D}_1$ of $N$ data points of dimension $D$ in general position, along with their associated labels. Let $\mathcal{S}_{\text{kcombs}}$ denote the set of all $D$-combinations with respect to dataset $\mathcal{D}$. Then we have following relation

$$\underset{s \in \mathcal{S}_{\text{kcombs}}(D, \mathcal{D})}{\text{argmin}} \min(E_{0\text{-}1}(\boldsymbol{w}_s, \mathcal{D}_1), E_{0\text{-}1}(-\boldsymbol{w}_s, \mathcal{D}_1)) \subseteq \underset{\boldsymbol{w} \in \mathbb{R}^{D+1}}{\text{argmin}}\, E_{0\text{-}1}(\boldsymbol{w}, \mathcal{D}_1) \tag{8}$$

where $\boldsymbol{w}_s$ represents the normal vector of the hyperplane that pass through the $D$-combination of data $s$, and $-\boldsymbol{w}_s$ is the negation of $\boldsymbol{w}_s$. The inner $\min$ on the left-hand side ensures that $s \in \mathcal{S}_{\text{kcombs}}\,(D, \mathcal{D})$ for each $s$, where $\mathcal{S}_{\text{kcombs}}\,(D, \mathcal{D})$ denote all possible $D$-combinations of the set $\mathcal{D}$. We take the smaller of $E_{0\text{-}1}\,(\boldsymbol{w}_s)$ and $E_{0\text{-}1}\,(-\boldsymbol{w}_s)$, and the outer argmin selects *one* of the values of that minimizes this quantity over all $s \in \mathcal{S}_{\text{kcombs}}\,(D, \mathcal{D})$.

## 2.4 Non-linear (polynomial hypersurface) classification

Based on the point-hyperplane duality, equivalence relations for linear classifiers on finite sets of data were established above. However, a linear classifier is often too restrictive in practice, as many problems require more complex decision boundaries. It is natural to ask whether it is possible to extend the theory to non-linear classification. This section examines a well-known concept in algebraic geometry, the *K-tuple Veronese embedding*, which allows the generalization of the previous strategy for solving classification problem with *hyperplane classifier* to problems involving *hypersurface classifiers*.

Importantly, we present the following theorem, which describes the relationship between hyperplane and hypersurface classification problems.

**Theorem 4.** *The K-tuple Veronese embedding.* Given variables $x_0, x_1, \ldots x_D$ in projective space $\mathbb{P}^D$ (which is isomorphic to the affine space $\mathbb{R}^D$ when ignoring the points at infinity (Cox et al., 1997)), let $M_0, M_1, \ldots M_G$ be all monomials of degree $K$ with variables $x_0, x_1, \ldots x_D$, where

---

**Algorithm 1** Incremental cell enumeration (ICE) algorithm

---

**Input**: $\mathcal{D}$: input dataset which consists of $N$ data points in $\mathbb{R}^D$ in general position; $\mathbf{l}$: label vector; $K$: degree of the polynomial;
**Output**: The optimal normal vector $\boldsymbol{w}^* : \mathbb{R}^{D+1}$ and optimal 0-1 loss $E_{0\text{-}1}^*$

1: $\mathcal{D}' = \rho_K(\mathcal{D})$ // *calculating embedded datasets*
2: $\boldsymbol{w}^* \leftarrow svm(\mathcal{D}'_\mathbf{l})$
3: $ds \leftarrow reorder(\boldsymbol{w}^*, \mathcal{D}')$ // *sort by* $|\boldsymbol{w}^\top \boldsymbol{x}|$
4: $Css \leftarrow [\,[\,],[\,],\ldots,[\,]\,]$ // $K{+}1$ *empty lists for 0 to K-combinations*
5: **for** $n = 0$ **to** $N-1$ **do**
6:     **for** $k = \min(K, n+1)$ **down to** $0$ **do**
7:         $Css[k] \leftarrow Css[k] \cup map(\lambda S.\ S \mathbin{+\!\!+} [n],\ Css[k-1])$ // *incremental combination generation*
8:     **end for**
9:     $\boldsymbol{ws} \leftarrow map(genModel(ds),\ Css[D])$ // *generate normal vectors from combinations*
10:     **for all** $\boldsymbol{w}' \in \boldsymbol{ws}$ **do**
11:         **if** $E_{0\text{-}1}(\boldsymbol{w}') \leq E_{0\text{-}1}(\boldsymbol{w}^*)$ **then**
12:             $\boldsymbol{w}^*, E_{0\text{-}1}^* \leftarrow \boldsymbol{w}', E_{0\text{-}1}(\boldsymbol{w}')$
13:         **end if**
14:         **if** $N - D - E_{0\text{-}1}(\boldsymbol{w}') \leq E_{0\text{-}1}(\boldsymbol{w}^*)$ **then**
15:             $\boldsymbol{w}^*, E_{0\text{-}1}^* \leftarrow -\boldsymbol{w}', N - D - E_{0\text{-}1}(\boldsymbol{w}')$ // *symmetric fusion law*
16:         **end if**
17:     **end for**
18:     $Css[D] \leftarrow [\,]$ // *eliminate D-combinations after use*
19: **end for**
20: **return** $\boldsymbol{w}^*, E_{0\text{-}1}^*$

---

$G = \begin{pmatrix} D+K \\ D \end{pmatrix} - 1$ (see Appendix A for the formal definition of monomials and polynomials and explanation of $G$). Define a mapping $\rho_K : \mathbb{P}^D \to \mathbb{P}^G$ which sends the point $\bar{\boldsymbol{p}} = (p_0, p_1, \ldots p_D) \in \mathbb{P}^D$ to the point $\rho_K(\bar{\boldsymbol{p}}) = (M_0(\bar{\boldsymbol{p}}), M_1(\bar{\boldsymbol{p}}), \ldots M_G(\bar{\boldsymbol{p}}))$. This is called the $K$-tuple Veronese embedding of $\mathbb{P}^D$ in $\mathbb{P}^G$. The hyperplane classification over the embedded datasets $\rho_K(\mathcal{D})$ is isomorphic to the polynomial hypersurface classification (defined by a degree $K$ polynomial) over the original dataset $\mathcal{D}$.

It is now straightforward to extend Theorem 3 to the following polynomial hypersurface classification theorem.

**Corollary 1.** *0-1 loss polynomial hypersurface classification theorem.* Consider a dataset $xs$ of $N$ data points in $\mathbb{R}^D$ in general position, along with their associated labels. Let $\rho_K(\mathcal{D})$ be the $K$-tuple Veronese embedding defined by monomials of degree $K$, we have following relation

$$\underset{s \in \mathcal{S}_{kcombs}(G, \rho_K(\mathcal{D}))}{argmin} \min(E_{0\text{-}1}(\boldsymbol{w}_s, \rho_K(\mathcal{D}_\mathbf{l})), E_{0\text{-}1}(-\boldsymbol{w}_s, \rho_K(\mathcal{D}_\mathbf{l}))) \subseteq \underset{\boldsymbol{w} \in \mathbb{R}^{G+1}}{argmin}\, E_{0\text{-}1}(\boldsymbol{w}, \rho_K(\mathcal{D}_\mathbf{l})) \quad (9)$$

where $\boldsymbol{w}_s \in \mathbb{R}^G$ denote as the normal vector determined by $s$ ($G$ data points).

## 2.5 Incremental cell enumeration (ICE) algorithm

Due to the *symmetry* of the 0-1 loss, where a data item is assigned a label of either $1$ or $-1$, the 0-1 loss for the negative orientation of a hyperplane can be directly derived from the positive orientation of the same hyperplane without calculating it explicitly. The following theorem formalizes this relationship.

**Theorem 5.** *Symmetry fusion theorem.* Consider a dataset $\mathcal{D}$ of $N$ data points of dimension $D$ in general position, along with their associated labels. Let $h$ be a hyperplane which goes through $D$ out of $N$ data points in the dataset $\mathcal{D}$, separating the dataset into two disjoint sets $\mathcal{D}^+$ and $\mathcal{D}^-$. If the 0-1 loss for the positive orientation of this hyperplane is $l$, then the 0-1 loss for the negative orientation of this hyperplane is $N - l - D$.

Therefore, the 0-1 loss linear classification problem can be solved by enumerating only the positive or negative-oriented hyperplanes, rather than both.

We now have all the necessary components to construct our algorithm, which enumerates all linear classification decision hyperplanes and thus solves (3). Theorem 3 states that all globally optimal solutions to this problem are equivalent (in terms of 0-1 loss) to the optimal solutions contained within the set of positive and negatively oriented linear classification decision hyperplanes (vertices

Table 1: Comparison of the accuracy of our novel ICE algorithm, against approximate methods on real-world datasets. Best performing algorithm is marked bold.

| dataset | $N$ | $D$ | ICE(%) | SVM(%) | LR(%) | LDA(%) |
|---------|-----|-----|--------|--------|-------|--------|
| HA | 283 | 3 | **77.03** | 72.08 | 73.14 | 73.85 |
| CA | 72 | 5 | **80.6** | 77.2 | 73.6 | 75.0 |
| CR | 89 | 6 | **95.51** | 91.10 | 89.89 | 89.89 |
| VP | 704 | 2 | **97.30** | 96.88 | 96.59 | 96.59 |
| BT | 502 | 4 | **78.69** | 74.50 | 75.50 | 74.10 |
| SP | 975 | 3 | **94.46** | 94.05 | 94.05 | 94.05 |

in the dual space) passing through $D$ out of $N$ data points in the dataset $\mathcal{D}$. There exist numerous algorithms for enumerating combinations; for example, Nguyen & Sanner (2013)'s PCS algorithm employed a one-by-one enumeration strategy. However, such a one-by-one approach is inefficient and unsuitable for optimization tasks, as it is non-recursive and therefore precludes the use of bounding methods for further acceleration.

He (2025) and He & Little (2025) provide an extensive discussion of various combination generators defined in both sequential and divide-and-conquer styles. We adopt the sequential generator introduced by He & Little (2025). The pseudocode is presented in Algorithm 1. The algorithm has a complexity of $O\left(N^{G+1} \times G^3\right)$, where $G$ is the dimension of the embedded space (with $G = D$ if $K = 1$). Since in line 18 of Algorithm 1 we eliminate $D$-combinations at every recursive step, the algorithm's memory usage is $O\left(N^G\right)$.

## 3 EMPIRICAL EXPERIMENTS

In this section, we analyze the performance of our ICE algorithm empirically. Our evaluation aims to test the following hypotheses: (a) the ICE algorithm consistently achieves the highest training accuracy among competing algorithms when allowing ICE to run to termination; (b) the solutions with significantly higher training accuracy (learned using the ICE algorithm) also achieve higher accuracy on the test datasets, and (c) the observed wall-clock runtime aligns with the worst-case time complexity analysis.

**Exact linear (hyperplane) classification**   We first compare our exact algorithm, ICE, against support vector machines (SVM)[2], logistic regression (LR), and linear discriminant analysis (LDA) on linear setting, using binary classification datasets from the UCI machine learning repository (Dua & Graff, 2019). As shown in Table 1, the ICE algorithm consistently finds solutions with lower 0-1 loss than approximate algorithms.

Due to space constraints, the results of the runtime complexity analysis and out-of-sample tests are presented in Appendix B. Figure 4 shows that the empirical wall-clock runtime agrees closely with the theoretical predictions. In Figure 5, we compare ICE against the state-of-the-art BnB algorithm by Nguyen & Sanner (2013) for solving the 0-1 LCP. Our empirical analysis demonstrates that Nguyen & Sanner (2013)'s algorithm exhibits *exponential complexity* in the worst-case.

Additionally, we also compare the performance of the ICE and BnB algorithms with that of a mixed-integer programming (MIP) solver for the 0–1 LCP, implemented in MATLAB using the GLPK solver, results shown in Figure (Figure 6.). These show that while the MIP solver is more efficient than BnB on small datasets, its performance is less predictable compared with ICE and BnB.

From (1), *we anticipate that exact solutions will not only achieve lower 0-1 loss on training datasets but are also more likely to generalize better*, yielding lower 0-1 loss on test datasets. To evaluate this hypothesis, Table 2 reports the out-of-sample performance of the ICE algorithm using 5-fold cross-validation, compared against approximate algorithms. The results indicate that training a linear model with substantially lower training error than the approximate algorithms also leads to stronger generalization in out-of-sample tests, thereby refuting the notion that the optimal solution necessarily overfits the data.

---

[2]We tuned the SVM hyperparameters using a standard coarse grid search, testing a set of widely spaced values (e.g., $[0.01, 0.1, 1, \ldots, 10000]$) on a logarithmic scale.

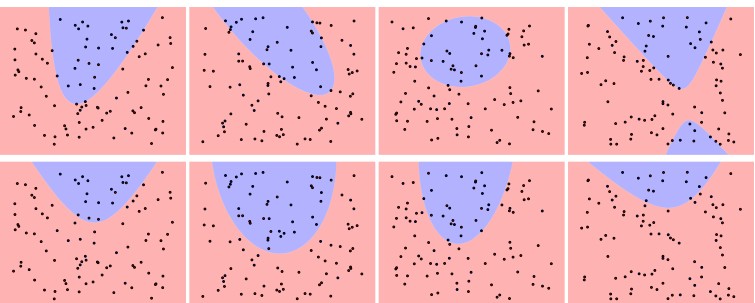

Figure 3: Optimal quadratic classifiers learned by the ICE algorithm (top four panels) achieve 0–1 losses of 9, 16, 17, and 16, while the approximate quadratic classifiers learned by an SVM with a degree-2 polynomial kernel (bottom four panels) obtain 0–1 losses of 17, 26, 21, and 22.

**Exact hypersurface (quadratic hypersurface) classification**    To evaluate the algorithm beyond linear classification, we test the ICE algorithm on synthetic datasets whose ground truth is a noisy quadratic boundary with label noise. We compute the exact solution (learned by ICE) on four datasets of size $N = 100$ and $D = 2$ and compare it against approximate solutions (learned by SVM with a degree-2 polynomial kernel). The results are shown in Figure 3. Similarly, the out-of-sample generalization performance on real-world datasets for the quadratic classifier is reported in Table 3.

## 4    SUMMARY, DISCUSSION AND FUTURE WORK

In this paper, we have presented *incremental cell enumeration*, ICE, the first provably correct, worst-case polynomial $O\left(N^{D+1}\right)$, which is polynomial in $N$ and exponential in varying $D$, run-time complexity algorithm for solving the 0-1 loss linear classification problem (3). Our empirical investigations show that the exact solution often significantly outperforms the best approximate solutions on the training dataset and also yields lower test error. This finding is critically important because it demonstrates that, contrary to widely held belief, globally optimal solutions to the 0-1 LCP can generalize well to unseen data. Prior to the development of ICE, provably correct exact algorithms—such as those proposed by Nguyen & Sanner (2013)—were computationally intractable even for moderate $N$ and small $D$, and their optimality had not been rigorously proved.

The immediate shortcoming of the algorithm is its exponential complexity in the data dimension $D$. This combinatorial complexity is further compounded in the hypersurface case, where the embedding space has dimension $O\left(D^K\right)$, resulting in a final hypersurface classification algorithm with time complexity $\left(N^{D^K}\right)$. However, since this problem is NP-hard, the exponential dependence on $D$ and $K$ is unlikely to be eliminated unless NP=P. Notably, the ICE algorithm relies solely on matrix operations, allowing for full vectorization and parallelization. Our current implementation uses simple parallelization via the PyTorch library. More sophisticated parallel implementations can be achieved by adopting the divide-and-conquer (D&C) combination generator introduced in (He & Little, 2025). A parallel implementation based on D&C-style recursion, executed on massively parallel GPUs, is expected to yield significantly better performance.

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

## A    PROOFS AND DEFINITIONS

**Definition 3.** *Monomial*. A *monomial* with respect to a $D$-tuple $\boldsymbol{x} = (x_1, x_2 \ldots, x_D)$ is a product of the form

$$M = \boldsymbol{x}^{\boldsymbol{\alpha}} = x_1^{\alpha_1} \cdot x_2^{\alpha_2} \ldots \cdot x_D^{\alpha_D}, \tag{10}$$

where $\boldsymbol{\alpha} = (\alpha_1, \alpha_2 \ldots, \alpha_D)$ and $\alpha_1, \alpha_2 \ldots, \alpha_D$ are nonnegative integers. The *total degree* of this monomial is the sum $|\boldsymbol{\alpha}| = \alpha_1 + \cdots + \alpha_n$. When $\boldsymbol{\alpha} = \boldsymbol{0} = (0, \ldots, 0)$, $\boldsymbol{x}^{\boldsymbol{0}} = 1$.

**Definition 4.** *Polynomial*. A polynomial $P$ in $x_1, x_2 \ldots, x_D$ with coefficients in $\mathbb{R}$ is a finite linear combination (with coefficients in the field $\mathbb{R}$) of monomials. A polynomial $P(\boldsymbol{x})$, or $P$ in short, will be given in the form

$$P(\boldsymbol{x}) = \sum_i w_i \boldsymbol{x}^{\boldsymbol{\alpha}_i}, w_i \in \mathbb{R}, \tag{11}$$

where $i$ is finite. The set of all polynomials with variables $x_1, x_2 \ldots, x_D$ and coefficients in $\mathbb{R}$ is denoted by $\mathbb{R}[x_1, x_2 \ldots, x_D]$ or $\mathbb{R}[\boldsymbol{x}]$.

Let $P = \sum_i w_i \boldsymbol{x}^{\boldsymbol{\alpha}_i}$ be a polynomial in $\mathbb{R}[\boldsymbol{x}]$. Then $\boldsymbol{\alpha}_i$ is called the *coefficient* of the monomial $\boldsymbol{x}^{\boldsymbol{\alpha}_i}$. If $w_i \neq 0$, then $w_i \boldsymbol{x}^{\boldsymbol{\alpha}_i}$ is called a *term* of $P$. The *maximal degree* of $P$, denoted $deg(P)$, is the maximum $|\boldsymbol{\alpha}_i|$ such that the coefficient $\alpha_i$ is nonzero. For instance, the polynomial $P(\boldsymbol{x}) = 5x_1^2 + 3x_1 x_2 + x_2^2 + x_1 + x_2 + 3$ for $\boldsymbol{x} \in \mathbb{R}^2$ has six terms and maximal degree two. Note that, since $|\boldsymbol{\alpha}_i|$ is defined as the sum of monomial degree, so polynomial $P'(\boldsymbol{x}) = 5x_1^2 x_2^2 + 3x_1 x_2$ has degree four.

The number of possible monomial terms of a degree $K$ polynomial is equivalent to the number of ways of selecting $K$ variables from the multisets of $D+1$ variables[3]. This is equivalent to the *size $K$ combinations of $D+1$ elements taken with replacement*. In other words, selecting $K$ variables from the variable set $(x_0, x_1, \ldots, x_D)$ in homogeneous coordinates with repetition, leads to the following fact.

**Fact 1.** If polynomial $P$ in $\mathbb{R}[x_1, x_2 \ldots, x_D]$ has maximal degree $K$, then polynomial $P$ has $\begin{pmatrix} D+K \\ D \end{pmatrix}$ monomial terms at most.

**Lemma.** For a set points $\mathcal{D} = \left\{ x_n \in \mathbb{R}^D : n \in \mathcal{N} \right\}$ in general position, the total number of linear dichotomies in Cover's function counting theorem, is the same as the number of cells of the dual arrangement $\mathcal{H}_{\mathcal{D}}$, plus the number of bounded cells of $\mathcal{H}_{\mathcal{D}}$. In other words, denote the number of dichotomies for $N$ data items in $\mathbb{R}^D$ as $\text{Cover}(N, D+1)$ ($D+1$ denote the dimension of data in

---

[3]There are $D+1$ variables considering polynomials in homogeneous coordinates, i.e. *projective space* $\mathbb{P}^D$.

homogeneous coordinates) and the number of cells and bounded cells of an hyperplane arrangement in $\mathbb{R}^D$ as $B_D(\mathcal{H}_\mathcal{D})$ and $C_D(\mathcal{H}_\mathcal{D})$. Then

$$\text{Cover}(N, D+1) = B_D(\mathcal{H}_\mathcal{D}) + C_D(\mathcal{H}_\mathcal{D}) \tag{12}$$

*Proof.* Given a set of points $\mathcal{D} = \left\{ x_n \in \mathbb{R}^D : n \in \mathcal{N} \right\}$ in general position. Cover, 1965's function counting theorem states that the number of linearly separable dichotomies given by affine hyperplanes is

$$\text{Cover}(N, D+1) = 2 \sum_{d=0}^{D} \binom{N-1}{d}. \tag{13}$$

The original Cover's function counting theorem counts the number of linearly separable dichotomies given by *linear* hyperplanes. However, the dual arrangement $\mathcal{H}_\mathcal{D}$ consists of a set of *affine* hyperplanes. Nevertheless, the number of dichotomies given by affine hyperplanes in $\mathbb{R}^D$ for dataset $\mathcal{D}$ is equivalent to the number of dichotomies given by linear hyperplanes for dataset $\bar{\mathcal{D}}$ in $\mathbb{R}^{D+1}$ (where $\bar{\mathcal{D}}$ is the *homogeneous dataset*, which is obtained by embedding $\mathcal{D}$ in homogeneous space. Recall that, $\bar{x} = (x, 1)$ is the data in homogeneous coordinates).

Alexanderson & Wetzel (1978) (and see also Edelsbrunner et al., 1986) show that shows that for a simple arrangement $\mathcal{H} = \{h_n : n \in \mathcal{N}\}$ in $\mathbb{R}^D$, the number of cells is $C_D(\mathcal{H}) = \sum_{d=0}^{D} \binom{N}{d}$, and the number of bounded regions is $B_D(\mathcal{H}) = \binom{N-1}{D}$.

Putting these two pieces of information together, obtains

$$\begin{aligned}
&B_D(\mathcal{H}_\mathcal{D}) + C_D(\mathcal{H}_\mathcal{D}) \\
&= \binom{N-1}{D} + \sum_{d=0}^{D} \binom{N}{d} \\
&= \binom{N-1}{D} + \sum_{d=0}^{D} \left[ \binom{N-1}{d} + \binom{N-1}{d-1} \right] \\
&= \sum_{d=0}^{D} \binom{N-1}{d} + \sum_{d=0}^{D} \binom{N-1}{d} \\
&= 2 \sum_{d=0}^{D} \binom{N-1}{d} \\
&= \text{Cover}(N, D+1).
\end{aligned} \tag{14}$$

$\square$

**Lemma.** For a dataset $\mathcal{D}$ in general position, each of Cover's dichotomies corresponds to a cell in the dual space, and dichotomies corresponding to bounded cells have no *complement cell* (cells with reverse sign vector). Dichotomies corresponding to the unbounded cells in the dual arrangements $\phi(\mathcal{D})$ have a complement cell.

*Proof.* The first statement is true because of the order preservation property – data item $x$ lies above (below) hyperplane $h$ if and only if point $\phi^{-1}(h)$ lies above (below) hyperplane $\phi(x)$. For a dataset $\mathcal{D}$ and hyperplane $h$, assume $h$ has a normal vector $w$ (in homogeneous coordinates) and there is no data item lying on $h$. Then, hyperplane $h$ will partition the set $\mathcal{D}$ into two subsets $\mathcal{D}_h^+ = \left\{ x_n : w^T x > 0 \right\}$ and $\mathcal{D}_h^- = \left\{ x_n : w^T x < 0 \right\}$, and according to the Thm. 1, $\mathcal{D}$ has a unique associated dual arrangement $\phi(\mathcal{D})$. Thus, the sign vector of the point $\phi^{-1}(h)$ with respect to arrangement $\phi(\mathcal{D})$ partitions the arrangement into two subsets $\phi_h(\mathcal{D})^+ = \left\{ \phi(x_n) : \nu_{\phi(x_n)}^T \phi^{-1}(h) > 0 \right\}$ and $\phi_h(\mathcal{D})^- = \left\{ \nu_{\phi(x_n)}^T \phi^{-1}(h) < 0 \right\}$, where $\nu_{\phi(x_n)}^T$ is the normal vector to the dual hyperplane $\phi(x_n)$, in other words, point $\phi^{-1}(h)$ lies in a cell of arrangement $\phi(\mathcal{D})$.

Next, it is necessary to prove that bounded cells have no complement cell. The reverse assignment of the bounded cells of the dual arrangements $\phi(\mathcal{D})$ cannot appear in the primal space since the transformation $\phi$ can only have normal vector $\boldsymbol{\nu}$ pointing in one direction, in other words, transformation $\phi$: $x_D = p_1 x_1 + p_2 x_2 + ... + p_{D-1} x_{D-1} - p_D$ implies the $D$th component of normal vector $\boldsymbol{\nu}$ is $-1$. For unbounded cells, in dual space, every unbounded cell $f$ associates with another cell $g$, such that $g$ has an opposite sign vector to $f$. This is because every hyperplane $\phi(\boldsymbol{x}_n)$ is cut by another $N-1$ hyperplanes into $N+1$ pieces (since in a simple arrangement no two hyperplanes are parallel), and each of the hyperplanes contains two rays, call them $\boldsymbol{r}_1, \boldsymbol{r}_2$. These two rays point in opposite directions, which means that the cell incident with $\boldsymbol{r}_1$ has an opposite sign vector to $\boldsymbol{r}_2$ with respect to all other $N-1$ hyperplanes. Therefore, it is only necessary to take the cell $f$ incident with $\boldsymbol{r}_1$, and in the positive direction with respect to $\phi(\boldsymbol{x}_n)$, take $g$ to be the cell incident with $\boldsymbol{r}_2$, and in the negative direction with respect to $\phi(\boldsymbol{x}_n)$. In this way, two unbounded cells $f$ and $g$ are obtained with opposite sign vectors. This means that, for point $\phi^{-1}(h)$ in these unbounded cells, this hyperplane $h$ partitions the dataset to $\mathcal{D}_h^+$ and $\mathcal{D}_h^-$, and it is possible to move the position of hyperplane $h$ in the primal space. So, there exists a new hyperplane $h'$ obtained by moving $h$, and it partitions the dataset to $\mathcal{D}_{h'}^+ = \mathcal{D}_h^-$ and $\mathcal{D}_{h'}^- = \mathcal{D}_h^+$. In other words, $h'$ has opposite assignment compared to hyperplane $h$. This corresponds, in the dual space, to moving a point $\phi^{-1}(h)$ inside the cell $f$, to cell $g$. For instance, in the simplest case, a hyperplane can be moved from left-most to the right-most to obtain an opposite assignment without changing the direction of the normal vector. $\qquad\square$

Since each of Cover's dichotomies corresponds to a cell in the dual space, and dichotomies corresponding to bounded cells have no complement cell (cells with reverse sign vector), lemma 2 demonstrates that all possible *Cover's dichotomies* of a given dataset $\mathcal{D}$ can be obtained by enumerating the cells of an arrangement and the complemented cells of the bounded cells. The enumeration of the complements of the bounded cells requires an additional process, as the bounded cells within the arrangement do not have complementary cells. This result directly leads to the following theorem.

**Lemma.** For a dataset $\mathcal{D}$ in general position, a hyperplane with $k$ data items lying on it, $0 \le k \le D$ correspond to a $(D-k)$-face in the dual arrangement $\mathcal{H}_\mathcal{D}$. Hyperplanes with $D$ points lying on it, correspond to vertices in the dual arrangement.

*Proof.* According to the incidence preservation property, $k$ data items lying on a hyperplane will intersect with $k$ hyperplanes, and the intersection of $k$ hyperplanes will create a $(D-k)$-dimensional space, which is a $(D-k)$-face, and the 0-faces are the *vertices* of the arrangement. $\qquad\square$

**Lemma.** Given a hyperplane arrangement $\mathcal{H} = \{h_n : n \in \mathcal{N}\}$, for an arbitrary maximal face (cell) $f$, the sign vector of $f$ is $\text{sign}_\mathcal{H}(f)$. For an arbitrary $(D-d)$-dimension face $g$, $0 < d \le D$, the number of different signs of $\text{sign}_\mathcal{H}(g)$ with respect to $\text{sign}_\mathcal{H}(f)$ is larger than or equal to $d$, where equality holds only when $g$ is conformal to $f$ ($g$ is a subface of $f$).

*Proof.* Denote the number of different signs of $\text{sign}_\mathcal{H}(g)$ with respect to $\text{sign}_\mathcal{H}(f)$ by $E_{0\text{-}1}(g)$. In a simple arrangement, the sign vector $\text{sign}_\mathcal{H}(f)$ of a cell $f$ has no zero signs, and a $(D-d)$-dimension face has $d$ zero signs. Thus the number of different signs of $\text{sign}_\mathcal{H}(g)$ with respect to $\text{sign}_\mathcal{H}(f)$ must be larger than or equal to $d$, i.e., $E_{0\text{-}1}(f) \ge d$. If $\text{sep}(f,g) = \emptyset$, then $E_{0\text{-}1}(g) = d$ according to the definition of $\text{sep}(f,g) = \emptyset$. In this case, $f, g$ are conformal. By contrast, if $f, g$ are not conformal, i.e. $\text{sep}(f,g) \ne \emptyset$, and assuming $|\text{sep}(f,g)| = C$, then according to the definition of the objective function and conformal faces, $E_{0\text{-}1}(\text{sign}_\mathcal{H}(f)) = d + C$. Hence, $E_{0\text{-}1}(\text{sign}_\mathcal{H}(g)) \ge d$, and equality holds only when $g$ is conformal to $f$. $\qquad\square$

**Theorem.** Consider a dataset $\mathcal{D}$ of $N$ data points of dimension $D$ in general position, along with their associated labels, denote $\mathcal{S}_{\text{kcombs}}$ as the set of all $D$-combinations with respect to dataset $\mathcal{D}$. Then we have following inequality

$$\underset{s \in \mathcal{S}_{\text{kcombs}}}{\arg\min} \min\left(E_{0\text{-}1}(\boldsymbol{w}_s), E_{0\text{-}1}(-\boldsymbol{w}_s)\right) \subseteq \underset{\boldsymbol{w} \in \mathbb{R}^{D+1}}{\arg\min} E_{0\text{-}1}(\boldsymbol{w}) \qquad (15)$$

where $\boldsymbol{w}_s$ represents the normal vector of the hyperplane that pass through the $D$-combination of data $s$, and $-\boldsymbol{w}_s$ is the negation of $\boldsymbol{w}_s$. The inner $\min$ on the left-hand side ensures that for each $s \in \mathcal{S}_{\text{kcombs}}$, we take the smaller of $E_{0\text{-}1}(\boldsymbol{w}_s)$ and $E_{0\text{-}1}(-\boldsymbol{w}_s)$, the outer $\arg\min$ finds *one* of the value of

that minimizes this quantity over all $s \in \mathcal{S}_{\text{kcombs}}$. In other words, (15) means that all globally optimal solutions to problem (3), are equivalent (in terms of 0-1 loss) to the optimal solutions contained in the set of solutions of all positive and negatively-oriented linear classification decision hyperplanes (vertices in the dual space) which go through $D$ out of $N$ data points in the dataset $\mathcal{D}$.

*Proof.* First, transform a dataset $\mathcal{D}$ to its dual arrangement. According to Lemma 2 and Lemma 3, each dichotomy has a corresponding dual cell and if the sign vectors for all possible cells in the dual arrangement and their reverse signs are evaluated, the optimal solution for the 0-1 loss classification problem can be obtained. Assume the optimal cell is $f$, it is required to prove that, one of the adjacent vertices for this cell is also the optimal vertex. Then, finding an optimal vertex is equivalent to finding an optimal cell since the optimal cell is one of the adjacent cells of this vertex. According to Lemma 5, any vertices that are non-conformal have corresponding 0-1 loss with respect to $\text{sign}_{\mathcal{H}}(f)$ which is strictly greater than $D$. Since $f$ is optimal, any sign vectors with larger sign difference (with respect to $\text{sign}_{\mathcal{H}}(f)$) will have larger 0-1 loss value (with respect to true label $t$). Therefore, vertices that are conformal to $f$ will have smaller 0-1 loss value, thus one can evaluate all vertices (and the reverse sign vector for these vertices) and choose the best one, which, according to Lemma 3, is equivalent to evaluating all possible positive and negatively-oriented linear classification decision hyperplanes and choosing one linear decision boundary with the smallest 0-1 loss value. $\square$

**Theorem 6.** *Symmetry fusion theorem.* Consider a dataset $\mathcal{D}$ of $N$ data points of dimension $D$ in general position, along with their associated labels. Let $h$ be a hyperplane which goes through $D$ out of $N$ data points in the dataset $\mathcal{D}$, separating the dataset into two disjoint sets $\mathcal{D}^+$ and $\mathcal{D}^-$. If the 0-1 loss for the positive orientation of this hyperplane is $l$, then the 0-1 loss for the negative orientation of this hyperplane is $N - l - D$.

*Proof.* Assume there are $m^+$ and $m^-$ data points misclassified in $\mathcal{D}^+$ and $\mathcal{D}^-$, then the 0-1 loss for $h$ equals $l = m^+ + m^-$. Denote the hyperplane $h$ with negative orientation as $h^-$. In the partition introduced by $h^-$, all correctly classified data by $h$ will be misclassified in $h^-$. Thus the 0-1 loss of $h^-$ is $|\mathcal{D}^+| - m^+ + |\mathcal{D}^-| - m^-$. Since $|\mathcal{D}^+| + |\mathcal{D}^-| = N - D$, we obtain the 0-1 loss for $h^-$ which is $N - D - l$. $\square$

## B ADDTIONAL EXPERIMENTS

### B.1 RUN-TIME COMPLEXITY ANALYSIS

We test the wall clock time of our novel ICE algorithm on four different synthetic datasets with dimension ranging from $1D$ to $4D$. The $1D$-dimensional dataset has data size ranging from $N = 1000$ to $60000$, the $2D$-dimensional ranges from $150$ to $2400$, $3D$-dimensional from $50$ to $500$, and $4D$-dimensional data ranging from $30$ to $200$. The worst-case predictions are well-matched empirically (see Figure 4).

All decision boundaries computed by exact algorithms entail the same, globally optimal 0-1 loss. Therefore, the only meaningful comparison between ICE and any other exact algorithms is in terms of time complexity. Here, we compare the wall-clock run time of our ICE algorithm with the exact *branch-and-bound* (BnB) algorithm of Nguyen & Sanner (2013). As a branch-and-bound algorithm, in the worst case it must test all possible assignments of data points to labels which requires an exponential number of computations, by comparison to ICE's worst case polynomial time complexity arising from the enumeration of dichotomies instead. Empirical computations confirm this reasoning (see Figure 5), predicting for instance that for the $N = 150$ data size with $D = 3$, ICE would take **1.2 seconds** worst-case whereas BnB would take approximately $10^{10}$ seconds (nearly **317 years**), demonstrating the clear superiority of our approach.

Additionally, we compare the performance of the ICE and BnB algorithms with that of a mixed-integer programming (MIP) solver for the 0-1 LCP, the results is shown in Figure 6. The results show that while the MIP solver is more efficient than BnB on small datasets, its performance is less predictable compared with ICE and BnB. This is highlighted by the fact that the number of sampling points explored by the MIP solver is smaller than those of BnB and ICE, as the MIP solver had not yet terminated to obtain the exact solution within our three-hour time limit.

| datasets | $N$ | $D$ | ICE (%) | SVM (%) | LR (%) | LDA (%) |
|---|---|---|---|---|---|---|
| HA | 283 | 3 | ***77.35/74.39** (0.58)/(2.66) | 72.21/71.23 (0.58)/(2.00) | 72.39/72.28 (0.92)/(0.15) | 73.10/74.74 (0.85)/(2.66) |
| CA | 72 | 5 | * **81.75/61.33** (2.66)/(5.58) | 71.93/60.00 (7.85)/(0816) | 76.49/58.67 (4.40)/(7.30) | 76.14/58.67 (6.75)/(7.30) |
| CR | 89 | 6 | ***95.49**/83.33 (1.18)/(7.86) | 92.11/**85.56** (1.89)/(10.09) | 90.99/82.22 (2.36)/(9.13) | 90.99/82.22 (2.36)/(12.67) |
| VP | 704 | 2 | ***96.93/97.59** (0.44)/(0.15) | 96.77/97.02 (0.49)/(2.32) | 96.02/96.03 (0.00)/(0.29) | 96.48/96.88 (0.63)/(2.28) |
| BT | 502 | 4 | ***79.50/74.06** (0.82)/(2.36) | 74.96/72.67 (0.82)/(2.85) | 76.06/73.27 (0.79)/(3.50) | 75.81/73.07 (1.09)/(3.53) |
| SP | 975 | 3 | ***94.49/94.15** (0.27)/(1.00) | 94.13/93.74 (0.33)/(1.33) | 94.13/93.74 (0.33)/(0.13) | 94.13/93.74 (0.33)/(1.33) |
| Ai4i | 10000 | 6 | **97.45/97.40** (0.10)/(0.36) | 96.62/96.57 (0.33)/(0.53) | 96.99/96.90 (0.10)/(0.44) | 97.00/96.75 (0.13)/(0.33) |
| AIDS | 2139 | 23 | **87.75/87.61** (1.09)/(1.12) | 86.84/86.49 (0.32)/(1.24) | 86.56/86.58 (0.19)/(1.23) | 85.71/84.90 (0.25)/(1.30) |
| AL | 243 | 14 | **98.45/98.36** (0.92)/(2.37) | 95.77/95.10 (0.89)/(3.05) | 96.18/95.51 (1.06)/(4.16) | 94.53/88.57 (0.53)/(4.21) |
| AV | 2043 | 7 | **88.94/88.31** (0.25)/(2.21) | 87.14/87.33 (0.41)/(1.64) | 86.94/87.04 (0.36)/(1.51) | 86.27/86.70 (0.42)/(1.39) |
| RC | 3810 | 7 | **93.86**/92.55 (0.29)/(1.05) | 92.83/**92.78** (0.18)/(0.78) | 92.86/92.81 (0.25)/(0.67) | 93.14/92.65 (0.17)/(0.59) |
| DB | 1146 | 19 | **79.48/79.74** (1.76)/(0.70) | 69.63/67.65 (0.01)/(0.03) | 70.57/69.39 (0.01)/(0.03) | 73.93/70.61 (0.01)/(0.02) |
| SO | 1941 | 27 | **77.70/76.04** (0.45)/(0.84) | 73.03/73.62 (0.01)/(0.01) | 72.78/72.96 (0.00)/(0.02) | 73.81/74.81 (0.01)/(0.03) |
| SS | 51433 | 3 | **86.58/86.68** (0.04)/(0.19) | 82.78/82.71 (0.00)/(0.00) | 79.69/79.65 (0.00)/(0.00) | 80.31/80.32 (0.00)/(0.00) |

Table 2: Comparison of the accuracy of our novel incremental cell enumeration (ICE) algorithm, against approximate methods: SVM, logistic regression (LR), and linear discriminant analysis (LDA) on real-world datasets. Results are reported as mean accuracy loss over training and test sets in the format: Training Error / Test Error (Standard Deviation: Train / Test). Exact solutions are marked with *, otherwise approximate, obtained using stochastic coreset selection for tractability purposes (C). Best performing algorithm is marked bold.

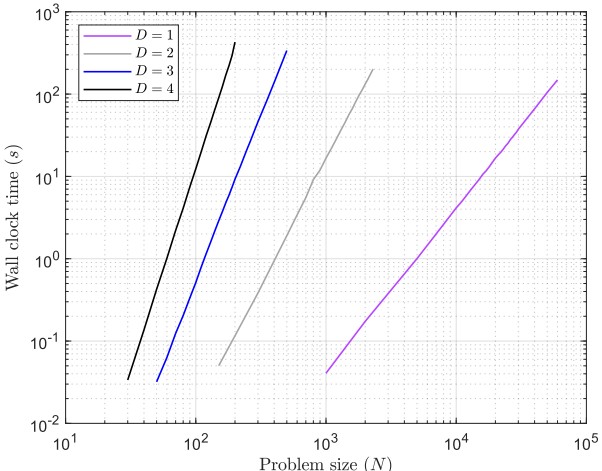

Figure 4: Log-log wall-clock run time (seconds) for the ICE algorithm in $1D$ to $4D$ synthetic datasets, against dataset size $N$, where the approximate upper bound is disabled (by setting it to $N$). The run-time curves from left to right (corresponding to $D = 1, 2, 3, 4$ respectively), have slopes 2.0, 3.1, 4.1, and 4.9, a very good match to the predicted worst-case run-time complexity of $O\left(N^2\right)$, $O\left(N^3\right)$, $O\left(N^4\right)$, and $O\left(N^5\right)$ respectively.

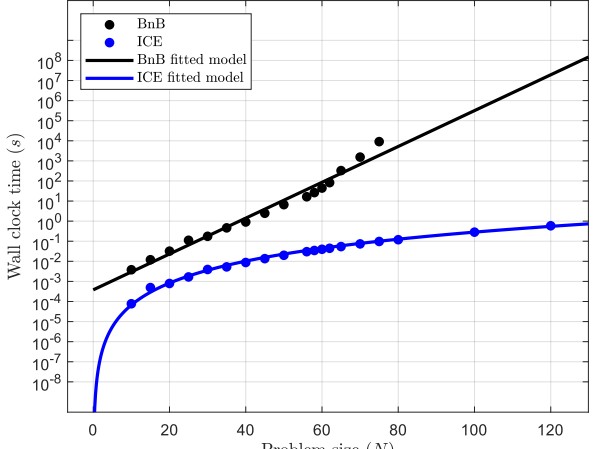

Figure 5: Log-linear wall-clock run time (seconds) plot comparing the ICE algorithm against the branch-and-bound (BnB) algorithm of Nguyen & Sanner (2013) (Matlab implementation provided by the authors) on three dimensional synthetic data. On this log-linear scale exponential run time appears as a linear function of problem size $N$, whereas, polynomial run time is a logarithmic function of $N$. Fitting appropriate models (lines) to the computational experiment data (dots) provides clear evidence of this prediction.

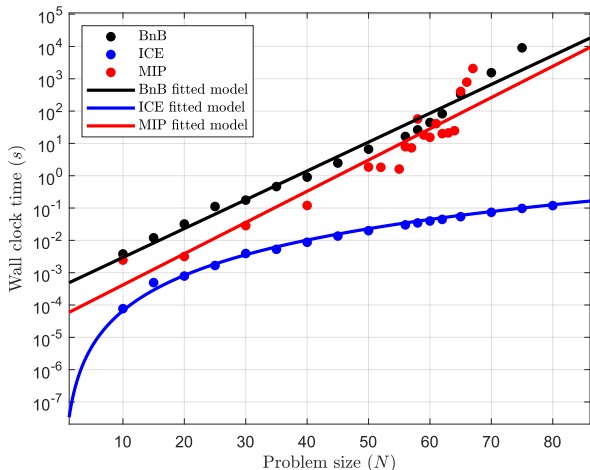

Figure 6: Log-linear wall-clock run time (seconds) plot comparing the ICE algorithm against the branch-and-bound (BnB) algorithm of Nguyen & Sanner (2013) (MATLAB implementation provided by the authors) and the mixed-integer programming (MIP) solver (implemented in MATLAB using GLPK solver) on three dimensional synthetic data. On this log-linear scale exponential run time appears as a linear function of problem size $N$, whereas, polynomial run time is a logarithmic function of $N$. Fitting appropriate models (lines) to the computational experiment data (dots) provides clear evidence of this prediction. The smaller sampling size of the MIP solver compared with BnB and ICE is due to the solver not terminating within the three-hour time limit, highlighting its much less predictable performance.

### B.2 OUT-OF-SAMPLE GENERALIZATION TEST

Due to the inherent combinatorial complexity of the 0-1 loss classification problem, the ICE algorithm becomes computationally intractable for high-dimensional datasets. In Table 2 , except for datasets that are tractable for ICE algorithm (those datasets evaluated in Table 1), we train all other datasets using the coreset selection method (ICE-coreset) introduced by (He et al., 2025). This method acts as a randomized wrapper for exact algorithms (pseudocode is provided in Appendix C). In brief, the coreset selection method reduces the dataset by eliminating subsets associated with solutions exhibiting higher 0-1 loss. This process is repeated until the reduced dataset (the coreset) becomes tractable for the ICE algorithm. As predicted, the solutions obtained by the ICE-coreset algorithm not only perform well on training data but also demonstrate higher test accuracy, refuting the misconception that exact algorithms necessarily overfit the training set.

### B.3 HYPERSURFACE CLASSIFICATION

The out-of-sample generalization performance on real-world datasets for the quadratic classifier is reported in Table 3.

### C CORESET SELECTION METHOD

Algorithm 2 shows the structure of the coreset selection method.

| datasets | $N$ | $D$ | ICE (%) | SVM (%) |
|---|---|---|---|---|
| HA | 283 | 3 | **77.35/73.68** (0.20/0.00) | 73.01/66.67 (0.00/0.00) |
| CA | 72 | 5 | **78.95/86.66** (0.01/0.00) | 70.18/46.67 (0.02/0.01) |
| CR | 89 | 6 | **92.96/94.44** 0.20/0.13 | 91.55/**94.44** (0.00/0.00) |
| VP | 704 | 2 | **97.12/96.45** (0.01/0.00) | 96.63/95.74 (0.00/0.01) |
| BT | 502 | 4 | **78.30/80.00** 0/0.44 | 73.56/75.25 (0.01/0.02) |
| SP | 975 | 3 | **95.13/91.79** (0.00/0.00) | 94.74/91.28 (0.00/0.00) |

Table 3: Empirical comparison of the training accuracy of our ICE algorithm against an approximate SVM with a degree-2 polynomial kernel on real-world datasets. Results are reported as mean accuracy over training and test sets in the format: Training Accuracy / Test Accuracy (Standard Deviation: Train / Test). Best performing algorithm is marked bold.

---

**Algorithm 2** ICE with coreset filtering

---

**Input**: $M$: block size; $R$: number of shuffles in each filtering round; $L$: max-heap size; $B_{\max}$: maximum input size for ICE algorithm; $c \in (0, 1]$: heap shrinking factor
**Output**: Max-heap $\mathcal{H}_L$ containing top $L$ configurations and associated data blocks

1: $\mathcal{C} \leftarrow ds$ // *initialize coreset with dataset*
2: **while** $|\mathcal{C}| \leq B_{\max}$ **do**
3:     Divide $\mathcal{C}$ into $\left\lceil \frac{|\mathcal{C}|}{M} \right\rceil$ blocks: $\mathcal{C}_B = \{C_1, C_2, \ldots, C_{\lceil \frac{|\mathcal{C}|}{M} \rceil}\}$
4:     Initialize max-heap $\mathcal{H}_L$ of size $L$
5:     **for** $r = 1$ **to** $R$ **do**
6:         **for all** $C \in \mathcal{C}_B$ **do**
7:             $cnfg \leftarrow ICE(\mathcal{D}_l, K)$
8:             $\mathcal{H}_L.\text{push}(cnfg, C)$
9:         **end for**
10:         $\mathcal{C} \leftarrow unique(\mathcal{H}_L)$ // *merge blocks and remove duplicates*
11:         $L \leftarrow L \times c$ // *shrink heap size*
12:     **end for**
13: **end while**
14: $cnfg \leftarrow ICE(\mathcal{D}_l, K)$ // *final refinement*
15: $\mathcal{H}_L.\text{push}(cnfg, \mathcal{C})$
16: **return** $\mathcal{H}_L$

---

