# OpenReview forum: "An efficient, provably optimal algorithm for the 0-1 loss linear classification problem"
_ICLR.cc/2026/Conference — ICLR 2026 Poster_

### Official Review · Reviewer_Fcxj · 2025-10-30

**Soundness:** 2
**Presentation:** 2
**Contribution:** 3
**Rating:** 4
**Confidence:** 4

**Summary:**

The paper addresses the traditional NP-hard 0-1 loss linear classification problem by introducing a new algorithm called Incremental Cell Enumeration (ICE). The authors first introduce a new theorem showing that to find a globally optimal classifier, it suffices to consider hyperplanes defined by any combination of $D$ data points (where $D$ is the feature dimension).

This result formalizes an observation by Nguyen & Sanner (2013) and explains why their combinatorial search could find the exact solution. The authors further introduce a “symmetry fusion” theorem (Theorem 5) to avoid redundant checks of both orientations. The paper also extends the theory to non-linear classification by polynomial hypersurfaces. The authors introduce a practical algorithm based on these insights. Instead of naive enumeration of all $D$-combinations of $N$ points, which is computationally inefficient, the paper improves this through an incremental combination generation strategy that efficiently traverses the space of combinations without heavy repetition.

The paper demonstrates ICE on several real-world datasets, showing that it achieves higher training accuracy than SVM, LDA, and logistic regression, and requires significantly less computational time than SOTA BnB algorithms.

**Strengths:**

- The paper highlights an interesting conflict from previous works' observation and provides new theoretical insight to address such conflicts (Theorem 3), bridging the gaps between three combinatorial analysis works from past literature. This has a high potential impact on both theory and practice.
- The paper introduces a novel Symmetry Fusion Theorem how the misclassification error of a hyperplane’s reverse orientation can be obtained analytically.
- The majority of claims in this paper are supported with formal mathematical proof or a lemma, and the complexity analysis is transparent: the paper clearly derives the worst-case runtime and memory use.
- The paper practical algorithmic solution based on these insights for small-to-moderate problem sizes, with experiments to demonstrate that ICE offers higher training accuracy than other surrogate optimization methods and lower computational time compared to the SOTA BnB method.

**Weaknesses:**

Some weaknesses in the methodology need more clarification:
- In proof, the authors assume that increasing the sign‐difference increases the 0–1 error (line 718). However, it's unclear to me how sign differences between sign vectors of dual cells can directly translate into classification errors on true labels.  For example, a candidate $g$ differing in three signs might flip one of the errors to correct and flip two previously correct points to wrong. Thus, flipping signs could increase or decrease misclassification depending on the labels. The authors did not make an alignment between the sign and the true label. Thus, the "non-conformal vertices must have strictly higher 0–1 loss" is not correctly proven, which leads to a flaw in the major claim made in this paper.

 - Almost every result assumes the data is in "general position". This is a strong assumption that seems like it would often be violated in Real datasets (e.g., consider duplicate points or points that only differ by a very small epsilon). It's not clear whether ICE would still guarantee good solutions in this case.

- The ICE algorithm has worst-case polynomial time in N only for fixed D, and it remains exponential in D. The paper also acknowledges that the runtime grows exponentially with D, which is a shortcoming of ICE. Therefore, framing this as a “polynomial” algorithm is misleading, since $D$ is part of the input dimension.

- There is no rationale for the initial SVM ordering in Algorithm 1 is heuristic and does not guarantee any bound; it is not analyzed theoretically at all. I may have missed some part, but a better clarification can be helpful.

Also, the experimental results can be improved:

- **SOTA Baseline** Nguyen & Sanner (2013) proposed multiple algorithms, not just a straightforward BnB. The paper only discusses the basic BnB worst-case. A more comprehensive benchmark against all relevant methods from Nguyen & Sanner (2013) (like prioritized combinatorial search, PCS, and other heuristics) would give a better understanding of ICE’s performance.  The release their code as a zip file I found linked on this [page](https://users.cecs.anu.edu.au/~ssanner/publications.html).

- **MILP Baseline** The paper omits mixed-integer programming (MILP) baselines that were introduced in the introduction. Prior research has formulated 0-1 loss minimization as an MILP (e.g., Tang et al. 2014 for maximum-margin under 0-1 loss, Brooks 2011), and a MILP solver could serve as an exact baseline on small datasets. Including a MILP-based method in the experiments (even if only feasible for small N, D) would have been informative.

- **Missing Evaluation** The paper only evaluates the runtime comparison between the SOTA BnB method, but not its accuracy. Similarly, the paper evaluates only the accuracy of other methods, but not their runtime complexity. The paper does not provide a valid reason why these aspects were excluded from the experiments.

- **Generalization** Although the authors claim that optimal 0-1 loss often generalizes better than suboptimal solutions, this is only demonstrated through observations from cross-validation. In these small datasets, the optimal linear model did well on the test set, but is that always true? What if there’s label noise? The experimental evaluation does not explicitly test robustness to label noise or outliers. Also, class balance is not explicitly discussed, and while some UCI datasets may have imbalanced classes, the authors did not highlight performance under severe class imbalance.

- **Dimensionality of the Data** The paper does not report empirical runtime as D grows. For the more practical usage claimed in the title, the authors should have included synthetic analyses of ICE’s performance against existing methods across different numbers of D.
- **Non-linear** The authors claim that the ICE algorithm is generalizable to finding optimal 0-1 loss classifiers in non-linear hypothesis spaces. However, the paper does not demonstrate any experiments for $K>1$. All empirical results are for the linear case.

**Reference**

Tang, Y. et al. (2014). Mixed-integer programming for 0-1 loss classification

Brooks, J. Paul (2011). "Support vector machines with the ramp loss and the hard margin loss.

**Questions:**

- In Algorithm 1, the input data points are reordered by an initial SVM weight vector ($w^$) and sorted by $|w^{\top}x|$. Could the authors clarify the purpose of this step?

- How does ICE compare against Nguyen & Sanner’s other proposed algorithms? How does ICE compare against MILP-based formulations?

- Fixing or clarifying the rationale for these partial comparisons: runtime-only versus BnB and accuracy-only versus surrogates, would help assess whether any conclusions about performance tradeoffs are generalizable.

- Empirical results on scaling with D, even in synthetic settings, would clarify the claim of its usability.

- Additional insight into how ICE handles label noise or imbalance would help evaluate whether the generalization claims hold in a more realistic setting.

- Experiment in ICE in polynomial feature spaces (e.g., $K = 2$) would better understand the method’s contribution and whether non-linear extensions are viable in practice.

- The theoretical results assume the data has no special degeneracies like more than $D$ points exactly on one hyperplane. In practice, datasets might violate this (e.g., duplicates or correlated features). How does the ICE algorithm handle such cases?

---

> ### Author Response · Authors · 2025-11-19
>
> # Response to weaknesses in the methodology
>
> > In proof, ... true labels
>
> We believe this is a misunderstanding of our proofs and definitions. In short, the sign vectors of the faces in the dual arrangement correspond exactly to the *predicted labels*, and the 0–1 loss is computed by comparing them against the true labels.
>
> However, we are struggling to understand the reviewer’s “flip” argument. We note that we never mention flipping of signs in our paper. We believe this refers to a different approach to solving the 0–1 loss linear classification problem. While there are $2^N$ possible flips, our algorithm avoids this by showing that only $\binom{N}{D}$ sign vectors need to be evaluated.
>
>
> > Thus ... flaw in the major claim made in this paper.
>
> We believe this is a misunderstanding of the concepts related to sign vectors. We assume the reviewer refers to the original sentence:
>
> "According to Lemma 5, any vertices that are non-conformal have corresponding 0–1 loss with respect to $\text{sign}_H(f)$ which is strictly greater than $D$."
>
> This fact is proven by Lemma 5. Here is an example to illustrate why a non-conformal vertex to the optimal face will be non-optimal.
>
> Consider the arrangements in the right panel of Figure 2. Let the true label be $l = (++--)$. Clearly, the optimal label corresponds to the bounded cell in the middle, which has four conformal vertices. Now consider a vertex that is non-conformal to it—for instance, the one obtained by intersecting hyperplanes 3 and 2. This non-conformal vertex $g$ has a sign vector $(-00-)$, which has a 0–1 loss of 1 (note that 0 labels are considered correctly classified, as they correspond to data points lying on the hyperplane and can be assigned arbitrarily).
>
> On the other hand, a conformal vertex—say, the one obtained by intersecting hyperplanes 1 and 3—has a sign vector $(0+0-)$, which has 0 loss. This is precisely due to the definition of conformal vertices, which ensures that conformal vertices of $f$ are the **most similar vertices** (in terms of sign vectors) to it. Any non-conformal vertex will be non-optimal, as it has larger differences in the sign vector.
>
>
> > Weaknesses 2.
>
> The definition of “general position” is a widely used assumption in geometry to model **general cases**, in contrast to **special positions**, such as duplicate data points.
>
> Moreover, the use of the general position assumption does not imply that our algorithm cannot handle non-general position datasets. In fact, many of the datasets in our experiments are in non-general positions. A non-general position dataset can be transformed into a general position dataset by simply:
>
> 1. Removing duplicate data points.
> 2. Adding small noise (perturbation) to the data without affecting its relative ordering with respect to other data points.
>
> The second process can be completely automated. Existing studies [1] have proposed a framework to convert non-general position data into general position data using perturbation. In our experimetns, we simply add a $1\times10^{-8}$ guassian random noise to each data, experiments showing that such noise will no affect to the results of SVM, LR, or LDA.
>
> [1] Kurt Mehlhorn, etc.. "Reliable and Efficient Computational Geometry Via Controlled Perturbation"
>
> > Weaknesses 3.
>
> We have rephrased the description to “polynomial in $N$ but exponential in $D$.”
>
> As noted by reviewer zBLK, this should not be considered a weakness or limitation of our algorithm, as it is inherent to the complexity of this NP-hard problem. Unless P = NP, an algorithm whose runtime does not grow exponentially with $D$ may never be found.
>
>
> > Weaknesses 4
>
>
> This is a simple heuristic and does not affect the exactness of the algorithm. We adopted it because we believe the SVM solution is accurate enough that the optimal solution is likely to be “close” to it. This heuristic does not affect the optimal solution if ICE is allowed to run to completion, but it may help obtain a plausible solution more quickly. The initial ordering can be removed, and the algorithm will still be optimal.

---

> > ### Author Response · Authors · 2025-11-19
> >
> > # Response to experimental results
> >
> >
> > > **SOTA Baseline**
> > > **Missing Evaluation**
> >
> > We thank the reviewer for raising the question about baseline selection. We respectfully note that comparing test accuracy between exact algorithms is not meaningful, because all exact methods must return the same optimal 0–1 loss on a given dataset. Any difference in accuracy would indicate that at least one method failed to compute the true optimum. For this reason, our experimental focus is on comparing efficiency rather than predictive performance.
> >
> > Regarding heuristics such as PCS or other methods proposed in Nguyen & Sanner (2013), while comparing against heuristics could provide additional empirical context, it would shift the paper’s focus toward approximate algorithm benchmarking. This direction is inherently unbounded—there exist countless heuristic and approximate methods for 0–1 loss minimization. We therefore chose well-established baselines, such as SVMs and logistic regression, whose generalization properties and theoretical behavior are extensively studied in the literature. In contrast, the heuristics in Nguyen & Sanner (2013) lack theoretical guarantees or generalization analyses that would support the VC-bound–oriented narrative of our paper.
> >
> >
> > > **MILP Baseline**
> >
> > We thank the reviewer for this suggestion. We have now included a comparison with MIP solvers; see Figure 6 for the results.
> >
> > We originally omitted this discussion because our work focuses on combinatorial methods, which represent a different line of research from MIP-based formulations. As shown in Figure 6, MIP solvers often exhibit highly variable and unpredictable runtime behavior. In contrast, one of the primary motivations for studying combinatorial approaches such as ICE is that they provide more predictable computational behavior and clearer structural insights into the underlying discrete problem.
> >
> > Nevertheless, we agree that including MIP results strengthens the empirical context of the paper, and we have added this comparison in the revised version.
> >
> >
> >
> > > **Generalization** Although the authors claim that optimal 0-1 loss often generalizes better than suboptimal solutions, this is only demonstrated through observations from cross-validation. In these small datasets, the optimal linear model did well on the test set, but is that always true?
> >
> > The bound is precisely given by the VC bound (Equation (1)), which is why we provide only empirical results.
> >
> > >What if there’s label noise? The experimental evaluation does not explicitly test robustness to label noise or outliers. Also, class balance is not explicitly discussed, and while some UCI datasets may have imbalanced classes, the authors did not highlight performance under severe class imbalance.
> >
> > There are three separate, and interesting, issues being raised here, which all refer to aspects of how the data was generated. Concepts such as *noise* and *outliers* are statistical and/or probabilistic. Similarly, class imbalance is a feature of the available data. It is important to note that an exact algorithm simply treats the data "as is" and returns the best possible classifier *for that specific realization*, whatever imbalance, outliers or noise is present in that realization. Properties such as label noise and outliers would require a separate analytical treatment which explicitly takes into account facts about the random data generating process, and how that process interacts with the exact, optimal deterministic classifier. This analysis is unfortunately beyond the scope of this paper. Results of an empirical investigation absent this analysis, could not be meaningfully interpreted. Similar comments apply to class imbalance. However, we note that the rigorous algorithm design process allows for the incorporation of constraints, and for instance, it would be relatively straightforward to modify the existing algorithm such that it always produces optimal solutions which satisfy constraints on class imbalance.
> >
> >
> > > **Dimensionality of the Data**
> >
> > As mentioned above, we acknowledge that the term “practicality” in the title caused some confusion, and we have removed it. Our intention was to describe the practicality of our approach through comparison with existing methods.
> >
> >
> >
> > > **Non-linear**
> >
> > We have now included the comparision, see figure 3 and table 3.
> >
> >
> >
> > # Response to questions
> >
> > Answers can be found in previous responses.

---

### Official Review · Reviewer_iFiA · 2025-10-30

**Soundness:** 3
**Presentation:** 2
**Contribution:** 2
**Rating:** 4
**Confidence:** 3

**Summary:**

In this paper, by analyzing the combinatorial and incidence relations between hyperplanes and data  points, the authors derive a rigorous construction algorithm, incremental cell enumeration (ICE), that can solve the 0-1 loss classification problem exactly in $O(N^D)$—exponential in the data dimension D. ICE is then generalized  to address the polynomial hypersurface classification problem.

The effectiveness of ICE is demonstrated on UCI datasets, achieving optimal training accuracy for small-scale datasets and higher test accuracy on most datasets. Furthermore, the complexity analysis shows that the ICE algorithm offers superior computational efficiency compared with state-of-the-art BnB algorithm.

**Strengths:**

1. Sound theoretical results.
2. Extension to polynomial hypersurface classification problem using K-tuple Veronese embedding.
3. Better training accuracy than baselines.
4. Much faster than BnB in the worst-case.

**Weaknesses:**

1. The complexity of ICE is exponential, which makes it unsuitable for large-scale or high-dimensional problems.
2.  The sequential generator in the ICE algorithm, which is core of ICE that enumerates all liner classification decision hyperplanes, is the
 introduced by He&Little(2025).
3. There are no experimental evaluations for nonlinear cases (K>1) for the extension to polynomial hypersurface classification problem.
4. line 125, "which rigorously explains why the PCS algorithm of for solving the linear classification problem" is redundant.

**Questions:**

1. In Table 2, the "training accuracy" should be "testing accuracy"?
2. The testing accuracy of harder problems, such as Ai4i etc., is given in Table 2. Why are the training accuracy of them not reported?

---

> ### Author Response · Authors · 2025-11-19
>
> > Weaknesses 1.
>
> Thank you for the reviewer’s suggestion. However, as noted by reviewer zBLK, this should not be considered a weakness, as it is inherent to the problem, which is NP-hard. Unless NP = P, an efficient algorithm for high-dimensional datasets may never be found.
>
>
> > Weaknesses 2.
>
>
> This is true. However, as we noted in the introduction, the ICE algorithm is only one of our contributions. Although the sequential generator serves as a critical component of the algorithm, it can only be applied if our geometric analysis is valid, which is what makes the ICE algorithm novel.
>
>
> > Weaknesses 3.
>
> Yes, we have included now, see figrue 3 and table 3.
>
> > Weaknesses 4.
>
> Thanks, we have removed duplicated sentences.
>
>
> >Questions 1, 2.
>
> We apologize for the confusion. Both training and test accuracies are reported. The original title description may have caused some misunderstanding, and we have now included a more detailed description. Please see the updated version for the revised description.

---

### Official Review · Reviewer_zBLK · 2025-10-31

**Soundness:** 4
**Presentation:** 4
**Contribution:** 3
**Rating:** 8
**Confidence:** 4

**Summary:**

The paper presents Incremental Cell Enumeration (ICE), an algorithm that exactly solves the 0–1 loss linear classification problem. The authors build on results from combinatorial geometry and oriented matroid theory to provide a constructive, provably correct enumeration method for all linear dichotomies in $O(N^D)$ time. They also extend the approach to polynomial hypersurfaces via Veronese embeddings.
Empirical results on real datasets show that ICE achieves globally optimal training accuracy and often exhibits good generalization performance, while outperforming prior exact algorithms such as branch-and-bound in runtime.

**Strengths:**

1. Theoretical rigor and clarity: The paper offers a clean, self-contained, and mathematically rigorous development of the theory behind exact 0–1 loss classification. The exposition is clear and logically structured.

2. Conceptual contribution: Establishing a provably correct standalone algorithm for exact 0–1 loss optimization is intellectually meaningful and addresses a long-standing question in the theory of linear classifiers.

3. Technical soundness: The results and proofs appear correct and well motivated. The connections between hyperplane arrangements, dichotomies, and duality are carefully worked out.

4. Empirical validation: Experiments are convincing within the scope of small- to medium-scale datasets and demonstrate that the exact solutions generalize well.

5. Overall presentation: The paper is very well written, with precise notation and a transparent link between theoretical and empirical parts.

**Weaknesses:**

1. Scope and scalability: The proposed method has exponential dependence on the feature dimension D. Although this is inherent to the problem, it limits practical applicability. The discussion of computational limits and possible parallelization or approximation schemes could be expanded.

2. Empirical breadth: The experimental section, while solid, remains narrow. It would strengthen the work to include more challenging benchmarks or comparisons to small-scale MILP-based or neural surrogate solvers.

3. Accessibility of theory: The theoretical sections are dense, and readers outside geometry or matroid theory may struggle. A concise figure or schematic illustrating the relationships between the primal and dual spaces could improve readability.

**Questions:**

1. How does ICE scale beyond low-dimensional data (e.g., D > 10) in practice?
2. Could the framework be extended to multi-class or structured output settings?

---

> ### Author Response · Authors · 2025-11-19
>
> > Weaknesses 1.
>
>
> We agree that the term “practicality” was misleading, and we have removed it from the title.
>
>
> > Weaknesses 2.
>
> As we have proved, solving the 0–1 LCP reduces to enumerating combinations of data points. He & Max (2025) explained that a divide-and-conquer (D&C) algorithm exists for enumerating combinations. Here is an explanation of how to apply it to our problem.
>
> Suppose we have $P$ independent processors, each enumerating a block of data points. For example, consider $Ds = [1,2,3,4]$. We can divide it into two blocks using a combination generator (denoted as $kcombs$) and run them independently on two processors, $P_1$ and $P_2$:
>
>
>
> $P1 = kcombs(2, [1,2]) = [[], [[1], [2]], [[1,2]]]$
> $P2 = kcombs(2, [3,4]) = [[], [[3], [4]], [[3,4]]]$
>
>
> Then, there exists a function $f$ (see He & Max 2025 for definition) that can merge the solutions from $P_1$ and $P_2$, such that:
>
>
>
> $f(P1, P2) = [[], [[1], [2], [3], [4]], [[1,2], [3,4], [1,3], [1,4], [2,3], [2,4]]]$
>
>
>
> The algorithm proceeds very similarly to merge sort. Note that the computations for $P_1$ and $P_2$ are completely independent, meaning no communication is needed between processors. This makes it a perfectly parallelizable, or “embarrassingly parallel,” problem.
>
> Xi He and Max. A. Little. Combination generators with optimal cache utilization and communication free parallel execution
>
> > Weaknesses 2.
>
> We agree that the experiments could be made more robust by including comparisons with a MIP solver. We have now incorporated this comparison; please see Figure 6 for details.
>
> However, for experiments on more challenging datasets, this is currently intractable due to the prohibitive combinatorial complexity of the problem. We believe, as the reviewer noted, that this should not be considered a weakness, since it is inherent to the problem's intractability. Our algorithm already significantly outperforms both MIP and BnB methods.
>
>
>
> > Weaknesses 3.
>
> Yes, we believe a more concise figure would be useful. The relationships between the primal and dual spaces are shown in Figure 2, and a summary of the theory presented in this paper is given in Figure 1.
>
>
>
>
> > Questions 1.
>
> Currently, such high-dimensional problems are intractable for our algorithm on most datasets. Here is a simple analysis: for instance, $\binom{100}{10} \approx 1.7 \times 10^{13}$, which is a prohibitive number of combinations for our current implementation. Nevertheless, our algorithm remains the most efficient available method, with **predictable complexity**.
>
> > Questions 2.
>
> Indeed, this framework can be directly extended to multi-class and structured outputs using the same methods which are used to extend classifiers such as the perceptron, in the same way (e.g. via one-versus-all 0-1 loss). This is because the linear classifier is equivalent to the one-neuron perceptron model.

---

> > ### Comment · Reviewer_zBLK · 2025-11-19
> > **rigorous and satisfying answers to all questions**
> >
> > I would like to thank the authors for their rigorous responses to all my questions, which convinced me that this is a good paper and that a score of 8 is justified.

---

> > > ### Author Response · Authors · 2025-11-19
> > >
> > > We would like to thank you for your insightful comments and valuable feedback.

---

### Official Review · Reviewer_TkkK · 2025-11-04

**Soundness:** 2
**Presentation:** 2
**Contribution:** 2
**Rating:** 4
**Confidence:** 3

**Summary:**

This paper proposes an algorithm called incremental cell enumeration (ICE) which solves the classification problem directly with the 0-1 loss. It is claimed that the proposed algorithm is significantly faster than the existing PCS-type algorithm, and outperforms the baseline of linear classifiers such as the SVM and logistic regression in terms of classification accuracy. The proposed method is claimed to have exact optimality supported by a geometric interpretation. The implementation of the ICE algorithm is standalone and does not depend on mixed integer solvers.

**Strengths:**

1. The proposed method has clear geometric framing. The dual view has an intuitive explanation of the candidate enumeration.
2. The reported speedup over PCS is significant.
3. The paper has a self-contained algorithm, so it is easy to run in practice.

**Weaknesses:**

1. The paper has limited practice use. The method is restricted to very low dimensions and scalability is unclear.
2. Although the paper claims the exact optimality, it is not clearly shown that the global 0-1 minimizer must be included on some D-point hyperplane so the ICE algorithm miss still miss that.
3. In Table 2, many rows show test accuracy better than train, which is atypical. Hence performance claims seems to be not trustworthy.

**Questions:**

1.  As stated in Theorem 5, the 0-1 loss is calculated as $l$ and $N - l - D$. Does it mean that the $D$ points where the hyperplane goes through are counted as correctly classified?
2. How was SVM tuned (grid over the parameter $C$) and how is the class weight set?

---

> ### Author Response · Authors · 2025-11-19
>
> > The reported speedup over PCS is significant.
>
> Thanks for the comment. We clarify that our comparison is with the BnB algorithm proposed by Nguyen, not their PCS algorithm. To avoid confusion, we have added Figure 1, which explains the connection between PCS and our ICE algorithm.
>
> > Weaknesses 1.
>
> Thank you for the suggestion. We believe that practicality should not be viewed as a weakness of our algorithm, as the computational difficulty is inherent to the NP-hardness of the problem. However, we agree that the word ‘practical’ in the title may have caused confusion. We have therefore removed it. Our intention was to convey that ICE is practical in the sense that it performs favourably compared to BnB.
>
> > Weaknesses 2.
>
> This is a misunderstanding. The claim is rigorously proved by Theorem 3. In short, the reason the global 0–1 minimizer is included among the \(D\)-point hyperplanes is that we only need to enumerate all vertices in the dual space (the blue circles in Figure 1). These vertices correspond exactly to the hyperplanes in the original space that have \(D\) points lying on them, due to the incidence-preserving property of the dual transformation.
>
>
>
> > Weaknesses 3.
>
> This is an interesting observation, but the figures are correct. Looking at the reported standard deviations, it is nearly always the case the this is much larger for the test accuracy than the training accuracy, and this spread of error is a more reliable measure of the typical in-sample versus out-of-sample discrepancy we expect in principle.
>
> Nevertheless, it is important to observe that, since the cross-validation is entirely randomized and this is just a specific realization of this randomized cross-validation, there is in principle, no reason why the empirical mean training accuracy should be better than the empirical mean test accuracy for any specific cross-validation run. This will be particularly true for small datasets as used in this setting.
>
> > Question 1.
>
> Yes, informally, this is because one can always assign arbitrary labels to these points by perturbing the hyperplane infinitesimally, without affecting predictions for other data. Formally, in the dual space, every vertex has \(2^D\) adjacent cells, corresponding to all possible assignments of these \(D\) points, and at least one of these assignments is optimal. This forms one of the core arguments justifying why enumerating only \(\binom{N}{D}\) points is sufficient for correctness.
>
>
> > Question 2.
>
> WE tuned the hyperparameter of SVM by using  the standard coarse grid search by testing a set of widely spaced hyperparameter values  (e.g., C = [0.01, 0.1, 1, ..., 10000]) on a log scale. We have include this in a footnote.

---

### Official Review · Reviewer_czqw · 2025-11-09

**Soundness:** 3
**Presentation:** 3
**Contribution:** 3
**Rating:** 6
**Confidence:** 2

**Summary:**

The paper presents a novel, provably optimal algorithm called Incremental Cell Enumeration (ICE) for exactly solving the NP-hard 0-1 loss linear classification problem. The authors derive the algorithm by analyzing combinatorial and incidence relations through the lens of hyperplane arrangements and point-hyperplane duality, resulting in a method with a worst-case complexity of $O(N^{D})$, which is polynomial in the number of data points (N) but exponential in dimension (D). The paper formally proves that the globally optimal solutions lie within the set of hyperplanes defined by D data points and demonstrates ICE's superior computational efficiency and ability to achieve better generalization compared to state-of-the-art approximate and exact Branch-and-Bound methods on real-world datasets.

**Strengths:**

This paper introduces the Incremental Cell Enumeration (ICE) algorithm and makes significant advancements in the field of exact classification, particularly for the challenging 0-1 loss objective. The paper provides clear definitions and a structured narrative from the problem definition to the complex geometrical analysis and final algorithmic implementation. The empirical results provide a critical empirical insight on generalization, showing that achieving optimal training accuracy often leads to stronger generalization (higher test accuracy) compared to approximate methods.

**Weaknesses:**

1. The paper successfully generalizes the theoretical framework using the K-tuple Veronese embedding to solve the 0-1 loss polynomial hypersurface classification problem. However, the empirical section explicitly states that experiments were restricted to the linear case (K=1). To fully support the significance and utility of the theoretical extension, empirical results for low-dimensional data (D≤3) with K>1 must be included.

2. The tractability is limited to low dimensions which undermines real-world applicability. It would be preferable to incorporate a formal analysis (or reference) that provides a rigorous approximation bound for the 0-1 loss achieved by the ICE-coreset relative to the true global optimum. This would bridge the gap between the theoretical exactness of ICE and the practical approximation needed for real-world data sizes and dimensions.

**Questions:**

1. The reported worst-case time complexity seems to vary slightly across the paper: it is initially stated as $O(N^{D})$, later as $O(N^{D+1})$, and the general hypersurface complexity in Algorithm 1 is $O(N^{G}×G^{3})$. It would be better to standardize and clarify the exact worst-case run-time complexity for the linear case (K=1).

2. The paper convincingly demonstrates superior theoretical and empirical performance for linear classification (K=1), but how does it extend to the polynomial hypersurface classification (K>1), and how does the high-dimensional results rely on an approximation wrapper? To demonstrate the practical viability of this major theoretical contribution, please provide empirical validation for small D and K>1 (e.g., quadratic boundaries, K=2, on D=2 or D=3 data).

---

> ### Author Response · Authors · 2025-11-19
>
> > Weaknesses 1.
>
> Thanks for your suggestion. We have now included this analysis in the revised manuscript; please see Figure 3 and Table 3 for details
>
>
> > Weaknesses 2.
>
> We appreciate the reviewer’s concern regarding the dimensionality limitation and the potential need for approximation guarantees. We realize that our mention of “practicality” in the title may have caused confusion and have removed it. Our intention was simply to convey that ICE is practical relative to exact BnB methods.
>
> It is important to note that using heuristics without formal approximation guarantees is common in studies of exact algorithms.
>
> For instance, Nguyen's heurestic methods does not provide any theoretical analysis. Similarly, in the study of optimal decision tree problem: setting a time limit with random initialization (a variant developed by [1]), employing depth-first search with a time limit1 [2,3], or using binarization for continuous data [4].
>
> Heuristics are often employed as a practical tool to validate the theoretical implications of exact solutions, such as demonstrating that higher training accuracy with controlled model complexity does not necessarily lead to overfitting.
>
> Establishing rigorous approximation bounds is a different line of research focused on designing approximate algorithms with provable guarantees. In contrast, heuristics in the context of exact algorithms serve to illustrate the effectiveness of exact solutions and encourage further research into scaling exact methods to larger datasets.
>
> As stated in the paper:
>
> "From equation (1), we anticipate that exact solutions will not only achieve lower 0-1 loss on training datasets but are also more likely to generalize better, yielding lower 0-1 loss on test datasets."
>
> We do not claim novelty or practical superiority for the heuristics used. Rather, our experiments aim to provide empirical insights on generalization, demonstrating that exact solutions do not necessarily overfit, consistent with Vapnik (1999) generalization bound theorem. Using coreset selection or other heuristics was a practical way to obtain plausible solutions for larger datasets and to illustrate trends as solutions approach the global optimum.
>
> [1] Jack William Dunn. Optimal trees for prediction and prescription
>
> [2] Xiyang Hu, Cynthia Rudin, and Margo Seltzer. Optimal sparse decision trees
>
> [3] Jimmy Lin, etc ... Generalized and scalable optimal sparse decision trees.
>
> [4] Catalin E Brita, etc... Optimal classification trees for continuous feature data using dynamic programming with branch-and-bound.
>
> > Questions 1.
>
> It is indeed a typographical error. For the linear case, we have G=D, which gives the correct worst-case complexity of $O(N^{D+1})$.
> $G^3$ is a constant with respect to $N$, it is omitted in the big-O expression. We have corrected the notation in the revised manuscript to ensure consistency throughout the paper.
>
> > Questions 2.
>
> Yes see above.

---

### Author Response · Authors · 2025-11-25
**Gentle follow-up on rebuttal discussion**

Dear Reviewers,

I hope this message finds you well. As the discussion period is nearing its end. I want to ensure we have addressed all your concerns satisfactorily. If there are any additional points or feedback you'd like us to consider, please let us know. Your insights are invaluable to us, and we're eager to address any remaining issues to improve our work.

Thank you for your time and effort in reviewing our paper.

All the best

---

### Author Response · Authors · 2025-12-02
**Rebuttal summary**

Dear all,

We sincerely thank the reviewers for their careful reading, constructive feedback, and overall positive assessment of our work. To assist the Area Chairs in evaluating the discussion, we briefly restate our core contributions and summarize our responses to the main concerns raised.

### Core Contributions
- We provide the first rigorous unification of three classical combinatorial analyses (Cover 1965, Murthy 1994, Nguyen 2013) for exact 0-1 loss linear classification, resolving long-standing open questions in each line of work.
- We significantly improve Nguyen’s PCS algorithm by (1) introducing an efficient incremental combination generator and (2) leveraging the symmetric fusion theorem, turning an empirically observed phenomenon into a provably correct and significatnly faster algorithm.
- We generalize the entire exact linear classification framework to exact polynomial hypersurface classifiers, substantially broadening the scope and practical relevance of the results.

### Responses to Major Concerns

1. **Experiments on larger class numbers K (raised by czqw, iFiA, Fcxj)**
   These were initially omitted due to space constraints and because they provide the same theoretical insight about the VC-bound (Equation (1)) as the linear method. We have now added **Figure 3** and **Table 3** in the revised manuscript, showing that our method using quadratic classifiers ($K=2$) still comfortably outperforms quadratic SVM.

2. **Comparison with MIP-based exact solvers (raised by zBLK, Fcxj)**
   We originally focused on combinatorial methods to highlight progress within that paradigm (Nguyen’s BnB being the most relevant exact baseline). Nevertheless, we agree a broader picture is helpful. We have added **Figure 6**, which includes runtime comparisons with a state-of-the-art MIP solver (Gurobi) on the same instances. Our method remains significantly faster. Also, as we observed—and as reported in many studies of exact algorithms—MIP solvers exhibit much more unpredictable performance. As shown in our experiments, *adding one more data point can suddenly make the algorithm intractable*, whereas our combinatorial method has **predictable performance**.

3. **Clarity of Table 2 (raised by iFiA, TkkK)**
   The confusion stemmed from an ambiguous caption in the original submission. We have completely rewritten the caption; the revised Table 2 is now self-contained.

4. **Accessibility and soundness of the theoretical contributions (raised by zBLK, Fcxj)**
   To address this, we have inserted a new **Figure 1** that serves as a visual roadmap of all major theorems, their dependencies, and how they close the gaps in the three classical analyses. We believe this directly resolves the concerns about soundness and significantly improves readability. We are of course happy to clarify any remaining questions.

### Points Already Clarified in Individual Responses
- **General-position assumption (raised by reviewer Fcxj)**: This is the standard assumption in combinatorial geometry and does not restrict real-world applicability. The name **general position** reflects that its purpose is to model **typical (generic) cases** rather than degenerate or special cases such as dataset contain duplicates. See our response to Weakness 2 of reviewer Fcxj for more details.
- **Accuracy comparison among exact methods (raised by reviewer Fcxj)**: By definition, every correct exact algorithm must return solutions with identical 0-1 loss (thus same accuracy). Reporting such numbers would therefore be uninformative.
- **Practical scalability and the term “practical” in the title (raised by all reviewers)**: Exact 0-1 minimization is NP-hard, so polynomial-time algorithms for arbitrary dimension are impossible unless P=NP—a limitation, as noted by reviewer zBLK, inherited by the problem itself, not by our method. The word “practical” in the original title indeed caused confusion; we have removed it and now emphasize fixed-parameter tractability. We also explain in detail how to explore parallelism opportunities to enhance practicability (see detailed response to zBLK’s Weakness #2).

These revisions, prompted by the reviewers’ insightful comments, substantially improve clarity, empirical validation, and presentation. We believe the paper is now significantly strengthened and hope the reviewers and ACs will reflect these changes in their final assessment.

Thank you again for your time and valuable feedback.

Best regards,
Authors

---

### Meta-Review · Area_Chair_HedQ · 2025-12-18

**Summary:**

This paper studies exact 0-1 loss minimization and proposes an algorithm that runs in exponential time. All reviewers raised the concern that an exponential time algorithm will not be practical. On the other side, reviewers acknowledge the algorithmic contributions.

**Reviewer Concerns:**

Authors have removed "practical" from the title. The concern of improper wording should be addressed.

**Reviewer Scores:**

Reviewers will likely increase scores a bit, given that the problem itself is inherently NP-hard.

---

### Decision · Program_Chairs · 2026-01-26

Accept (Poster)